

# Surveying Methane Point-Source Super-Emissions across Oil and Gas Basins with MethaneSAT

Luis Guanter[1,2], Javier Roger[2], Jack Warren[3], Maryann Sargent[4], Zhan Zhang[4], Sébastien Roche[3,4], Christopher Chan Miller[3,4], Michael Steiner[1], Harvey Hadfield[1], Mark Omara[3], James Williams[3], Katlyn MacKay[3], Jonathan E. Franklin[4], Steven C. Wofsy[4], Steven P. Hamburg[3], and Ritesh Gautam[3]

[1]Environmental Defense Fund, Amsterdam, The Netherlands
[2]Research Institute of Water and Environmental Engineering (IIAMA), Universitat Politècnica de València, València, Spain
[3]Environmental Defense Fund, New York, New York, USA
[4]Harvard John A. Paulson School of Engineering and Applied Sciences, Harvard University, Cambridge, MA, USA

**Correspondence:** Luis Guanter
(lguanter@edf.org)

**Abstract.**

Methane emissions from the oil and gas (O&G) industry play a major role in the global methane budget. The MethaneSAT space-based mission, which operated between March 2024 and June 2025, was designed to provide high-quality data on O&G methane emissions, including both regional fluxes and high-emitting point sources. This is enabled by MethaneSAT's high spectral resolution (0.25 nm), medium spatial sampling ($110 \times 400\,\mathrm{m^2}$ at nadir), and wide-area coverage (about 200 km at nadir). In this work, we showcase the potential of MethaneSAT to survey high-emitting point sources across O&G basins. We first assess MethaneSAT's performance for the detection, quantification and attribution of methane plumes through the analysis of key observation-related parameters, including wind speed, surface albedo and spatial sampling. We estimate a detection limit of about 500 kg/h for favourable observation conditions, which are mostly facilitated by low winds. We then analyse selected MethaneSAT datasets from the main O&G methane hotspots in the world. We observe particularly strong and persistent sources in the well-known Turkmenistan's South Caspian basin and the U.S. Permian Basin (especially across the Midland sub-basin), and reveal major super-emissions in the Maturin (Venezuela) and Zagros Foldbelt and Widyan (Iran) O&G basins, and the Appalachian basin including O&G and coal production. We also highlight other examples of strong methane sources at high latitudes (West Siberia), in offshore platforms in the Gulf of Mexico, and from the waste sector. Our results illustrate the potential of the MethaneSAT data archive for the discovery of new methane hotspot regions and super-emitters around the planet.

## 1 Introduction

Methane ($CH_4$) is a short-lived but powerful greenhouse gas, responsible for approximately a third of the global warming observed since preindustrial times (IPCC, 2021). The oil and gas (O&G) sector is a major contributor to anthropogenic methane emissions globally (Saunois et al., 2025). A substantial share of O&G methane emissions are caused by high-emitting point sources, also known as super-emitters, including leaks, venting, and incomplete combustion at flares (Zavala-Araiza et al.,





2017). These emissions are very often preventable. The O&G sector represents therefore a cost-effective and immediate target for emission reduction.

Satellite remote sensing has rapidly expanded our ability to operationally detect and quantify methane super-emitters around the globe. For example, the International Methane Emissions Observatory (IMEO) is compiling a satellite-based publicly-available database of methane plumes and local enhancements, which has become an important asset in our understanding of global point-source methane emissions (UNEP-IMEO, 2023). Despite these advances, however, current methane-sensitive satellite systems trade off between detection sensitivity, spatial resolution, and regional coverage, compromising their ability to simultaneously assess both high-emitting point sources and diffuse emissions at basin scale (Jacob et al., 2022).

On the one hand, the Sentinel-5P/TROPOMI mission offers global daily revisit at the expense of a relatively coarse spatial resolution of $7 \times 5.5 \, \text{km}^2$. TROPOMI is being used for the estimation of total regional fluxes through the inversion of atmospheric transport models (e.g. Zhang et al., 2020b; Varon et al., 2023). TROPOMI has also potential for the systematic detection and monitoring of very large point sources (Lauvaux et al., 2022; Schuit et al., 2023) as well as to identify regions with persistently-enhanced methane concentrations (Vanselow et al., 2024). However, the detection and attribution of single plumes with TROPOMI is limited by the mission's relatively coarse spatial sampling.

On the other hand, an increasing number of high-resolution satellite instruments, also known as point-source imagers, are being used for the detection and quantification of single point sources (Jacob et al., 2022). These include imaging spectrometers such as GHGSat, PRISMA, EnMAP, EMIT, and Carbon Mapper's Tanager-1 (Varon et al., 2020; Guanter et al., 2021; Roger et al., 2024b; Thorpe et al., 2023; Duren et al., 2025) as well as multispectral missions such as Sentinel-2 and Landsat (Varon et al., 2021; Gorroño et al., 2023; Vaughan et al., 2024). These missions have enabled large-scale surveys of methane point sources across different O&G basins, including the Permian-Delaware sub-basin (Irakulis-Loitxate et al., 2021) and Turkmenistan's West coast (Irakulis-Loitxate et al., 2022b), which are arguably two of the regions with the highest concentration of high-emitting point sources around the world. Conversely, these methane-sensitive high-resolution missions have low or no sensitivity to the diffuse component of the emissions and the overall background methane gradients. Also, they have a limited spatial coverage, either by design (the case of imaging spectrometers), or by the fact that the plume detection process can in general only be applied over pre-selected areas or sources (multispectral instruments) (Vaughan et al., 2024). This causes that multiple overpasses of these missions, sometimes over long time periods, are needed to characterize the emissions from an entire region, where only individual plumes are sampled.

MethaneSAT, a satellite mission developed by the Environmental Defense Fund and international partners, was designed to fill the gap in coverage and sensitivity that we find in the current landscape of methane-sensitive missions. MethaneSAT's ability to monitor 200-km swaths at medium spatial resolution and with a high spectral sensitivity allows to link individual sources to broader regional trends in methane concentration gradients, thereby enhancing the monitoring of sectoral emissions and informing mitigation efforts. It also enables the identification of large-area super-emitting regions, not just isolated point sources, within entire oil & gas basins.





In this work, we first evaluate the potential and limitations of MethaneSAT for the detection and quantification of methane plumes from point sources. Second, we exploit the existing archive of MethaneSAT observations over O&G basins between May 2024 and June 2025 to investigate methane super-emissions in known methane emission hotspots around the world.

## 2 Materials and Methods

### 2.1 The MethaneSAT mission and its data products

The MethaneSAT mission was designed to generate high-quality data on methane emissions around the world, with a focus on the O&G and agricultural sectors. It was launched on 4 March 2024, and stopped its operation on 20 June 2025 after contact with the satellite was lost. MethaneSAT consisted in a dual spectrometer system measuring in the 1249–1305 and 1598–1683 nm spectral windows with a spectral resolution (sampling) of 0.25 nm (0.08 nm). During observations from nadir, a 200-km swath is covered by $2048\times560$ pixels, with a rectangular pixel size of 110 m (across-track) and 400 m (along-track) (Environmental Defense Fund, 2021).

This instrument configuration enables the generation of accurate methane concentration maps (namely, dry column methane mixing ratio, $XCH_4$) from spectral radiance data cubes, which are MethaneSAT's Level-1 (L1) data product (Conway et al., 2024). The $CO_2$-proxy $XCH_4$ retrieval, described in Chan Miller et al. (2024), is used to generate $XCH_4$ maps at the native sensor coordinates (L2 product). The L2 product is converted in a L3 product after oversampling on a regular latitude/longitude grid, following the approach by Sun et al. (2018). $XCH_4$ maps are subsequently converted into methane emission data, which represents the mission's L4 product. This final product is split into total regional emission data (spatially-distributed emission rates across the >200-km area at a 1-km sampling) and point source emissions (coordinates and emission rates of the methane plumes detected from each acquisition). The latter is referred to as L4-plume product hereinafter in this work. Point-source emission rates are estimated from the detected plumes using an evolution of the methods described in Chulakadabba et al. (2023) for MethaneAIR, which is MethaneSAT's airborne precursor. These methods rely on the application of mass-balance models on single methane plumes for the estimation of emission rates from individual sources.

An example of a $XCH_4$ map from a MethaneSAT L2 $XCH_4$ product and the corresponding L4-plume dataset are represented in Fig. 1. A subset of a TROPOMI $XCH_4$ map from a dataset acquired within 70 min of the MethaneSAT acquisition is also represented in order to contextualize the different spatial sampling of the two missions. The details of this acquisition will be further discussed in Sect. 3.1.1.

### 2.2 Analysis of MethaneSAT's potential for methane plume detection

We have evaluated the impact of different observation-related factors on plume detection, including surface albedo, retrieval noise, pixel size, and wind speed. We combine these factors into the mass-balance model proposed by Jacob et al. (2016), which has also been used in other studies (MacLean et al., 2024; Duren et al., 2025). This model expresses the minimum



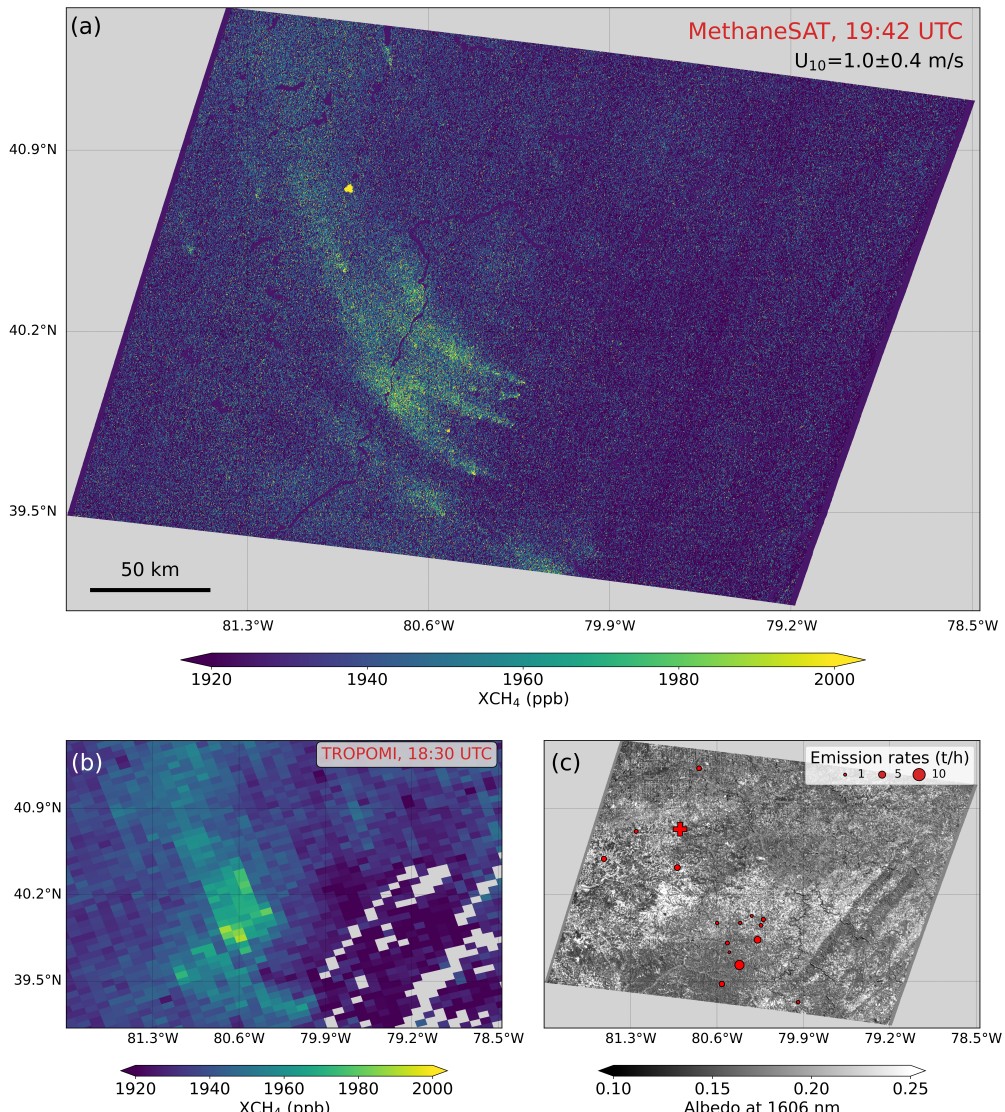

**Figure 1.** Maps of methane concentration (namely, dry column methane mixing ratio, $XCH_4$) obtained from (a) MethaneSAT and (b) TROPOMI overpasses over the Appalachian basin (USA) on 5 September 2024. The same color scale is used for both maps. The two acquisitions happened within a time difference of about 70 min. The 1606-nm albedo map from the MethaneSAT acquisition is displayed in (c). It also shows the location and emission rates quantified from the MethaneSAT level-4 plume product for this acquisition. The red cross in (c) depicts a short-lived emission with a flux rate higher than 100 t/h.

85   detection limit ($Q_{min}$), as

$$Q_{min} = n_p \frac{M_{CH_4}}{M_a} \frac{UWpq\sigma}{g}, \qquad (1)$$



where $M_a$=0.029 kg mol$^{-1}$ and $M_{CH_4}$=0.016 kg mol$^{-1}$ are the molecular weights of dry air and methane, $p$ is the dry atmospheric surface pressure, $g$ is the acceleration of gravity, $U$ is the wind speed, $W$ is the pixel size and $q$ is the number of standard deviations above the noise level required to reliably assign a methane enhancement to a pixel. Jacob et al. (2016) suggested to take $q = 2$ for plume detection. We have added the $n_p$ factor to the original model to represent the number of pixels $q$ times above the noise level that would be needed for the plume to be detected. We choose $n_p = 3$, i.e. we need to have three pixels with detectable enhancements for the whole plume to be detected. We acknowledge that the $q$ and $n_p$ parameters in this model are arbitrarily selected, and also that the $n_p$ factor is a simplistic representation of the plume detection process (e.g. a factor modeling the connectivity of several pixels would be more realistic). For this reason, we do not interpret $Q_{min}$ as an absolute measure of the minimum detectable emission rate, but only use it for comparative analysis, This includes the investigation of the relative impact of the different variables driving the plume detection process, and the intercomparison of the plume detection potential among different MethaneSAT L2 datasets.

In addition, we have run a first analysis of absolute plume detection limits using plumes generated by large-eddy simulations with the Weather and Research Forecasting Model (WRF-LES). WRF-LES plumes have been generated using different flux rates and wind speeds as inputs. The plumes were simulated at 30 m×30 m sampling, and then spatially-resampled to MethaneSAT's 100×400 m$^2$ spatial sampling. The spatially-distributed per-pixel methane concentration enhancement ($\Delta$XCH$_4$) maps from these WRF-LES plumes were added to real MethaneSAT XCH$_4$ maps, which resulted in hybrid real-simulated datasets, which include plumes with known $\Delta$XCH$_4$ values but preserve the noise and artifacts in the original data.

Plume simulations have also been used in this study to assess the impact of MethaneSAT's spatial sampling on the detection and attribution of methane plumes. WRF-LES plumes sampled at 30-m have been convolved with MethaneSAT's 110×400 m$^2$ for this purpose.

## 2.3 Matched-filter ΔXCH$_4$ retrievals

The CO$_2$-proxy retrieval used to generate MethaneSAT's official L2 products has been proven to enable accurate and robust XCH$_4$ retrievals, which are key to both the inversion of regional fluxes and the detection and quantification of point source emissions. On the other hand, the work by Guanter et al. (2025) with MethaneAIR suggested that the matched-filter data-driven retrieval can help improve the plume detection limits by increasing the retrieval precision with respect to the CO$_2$-proxy retrieval. Although the focus of this study is the analysis of the official MethaneSAT's L2 XCH$_4$ data product, we have also evaluated the potential of a matched-filter retrieval for an improved detection of point sources with MethaneSAT.

The matched-filter method is a data-driven retrieval that estimates $\Delta$XCH$_4$ from spectral radiance hypercubes. Details of the implementation of such retrieval for MethaneAIR, MethaneSAT's airborne precursor, are provided in Guanter et al. (2025). The $\Delta$XCH$_4$ maps derived with the matched-filter retrieval were shown to have a higher-signal-to-noise ratio than the ones produced from the official L2 CO$_2$-proxy retrieval. On the other hand, because of the matched-filter's statistical nature, its performance is affected by the characteristics of the input scene, including the spatial variability of the spectral radiance and the number of pixels available for the calculation of the mean spectrum and the covariance matrix which are at the core of the matched-filter. It is unclear how this scene-dependence affects the performance of the matched filter for MethaneSAT.



Our MethaneSAT implementation of the matched-filter is run in the 1599–1680 nm spectral fitting window. Similar to the MethaneAIR version, the retrieval is run column-wise in order to account for potential non-uniformities in the detector response and across-track variations in the viewing angle, which can be important for MethaneSAT.

## 2.4 MethaneSAT data used in this study

The main sources of input data for this work have been MethaneSAT L2 and L4-plume files (i.e. $XCH_4$ maps and the corresponding point-source product, respectively) from a number of targets, most of them sampling major O&G basins.

In particular, we have selected those sites where frequent super-emissions were detected with the TROPOMI mission according to the global database of methane plumes generated by the IMEO (United Nations Environment Programme, 2025), which is also a key dataset for this study. The spatial distribution of the selected sites is shown in Fig. 2. The regions inferred
from TROPOMI plume detections include the Permian Basin and the Haynesville-Bossier basin in the USA, the South Caspian and Amu Darya basins in Turkmenistan, the Hassi-Massoud basin in Algeria, and the Zagros-Foldbelt and Widyan basins in Iran. In addition to these sites, we have added other targets, including the Appalachian region in the USA and the Maturin basin in Venezuela, to illustrate different plume detection aspects of MethaneSAT.

For some of the $XCH_4$ map examples shown in this study, the $XCH_4$ enhancement ($\Delta XCH_4$) map, rather than the absolute
$XCH_4$, is shown. $\Delta XCH_4$ is calculated from the L2 $XCH_4$ data by subtracting the background $XCH_4$ level, which is estimated as the median $XCH_4$ of the scene. Non-valid pixels caused by missing data or retrieval artifacts are set to the average $XCH_4$ or $\Delta XCH_4$ values in the scene in order to facilitate the visualization of plumes. This is justified by the expected sparsity of plume pixels with respect to the total number of pixels in the scene.

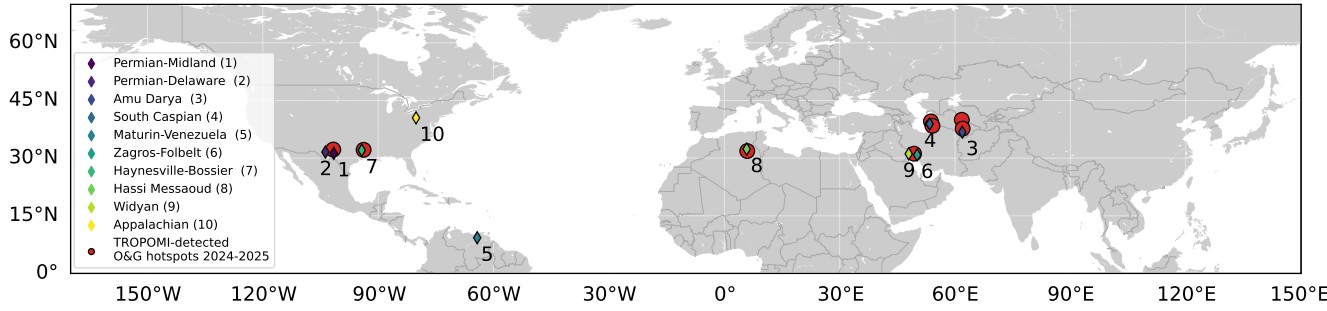

**Figure 2.** Main MethaneSAT oil and gas targets featured in this study, compared with areas with persistently-high methane emissions according to the TROPOMI data included in the IMEO plume database. The MethaneSAT targets are marked by coloured diamonds. The regions derived from the TROPOMI plume detections, marked by red circles, correspond to the $2°$ gridboxes where more than 5 methane enhancements have been reported from TROPOMI observations in 2024 and 2025.

MethaneSAT's L2 datasets include rasters of zonal and meridional 10-m wind from the GEOS-FP reanalysis product (Molod
et al., 2012), which has been used throughout this study for the analysis and interpretation of results.



## 3 Results

### 3.1 Evaluation of MethaneSAT's potential for the detection and attribution of methane plumes

#### 3.1.1 Information content in MethaneSAT XCH$_4$ maps

The data contained in MethaneSAT L2 and L4-plume files is illustrated in Fig. 1, which was briefly discussed in Sect. 2.1.

A more in-depth analysis of this acquisition is provided here. Fig. 1a shows a XCH$_4$ map derived from MethaneSAT over an area of about $300 \times 300$ km$^2$ at the central part of the Appalachian basin (USA). The equivalent map derived from a TROPOMI acquisition over the same area about 70 min earlier than the MethaneSAT overpass is shown in Fig. 1b. The visual comparison of the two XCH$_4$ maps confirms the consistency between the two datasets, and the higher level of detail enabled by MethaneSAT's higher spatial resolution ($110 \times 400$ m$^2$ for MethaneSAT, and $7 \times 5.5$ km$^2$ for TROPOMI). In addition, a number

of methane plumes and methane concentration enhancements can be seen across the scene. The plumes that were detected and quantified from this MethaneSAT dataset (i.e. the L4-plume product) are indicated in Fig. 1c on top of a map of near-infrared suface albedo derived from the same dataset. It comprises a total of 16 plumes, with emission rates ranging between roughly 800 and 7000 kg/h (see Fig. A1). These emissions from the Appalachian region are due to a combination of O&G, coal mining and waste facilities across the area, with the last two types of sources including a number of highly persistent point sources

(Warren et al., 2024). A massive emission of more than 100 metric tonnes per hour (t/h) was also detected (marked by red cross in Fig. 1c), but we assume it was short-lived because we were not able to detect it in a follow-up study using geostationary observations by the GOES-ABI mission, with a continuous coverage over North America every 5–10 min (Watine-Guiu et al., 2023).

#### 3.1.2 Assessment of plume detection limits

The lower end of the Appalachian point source emissions (about 500 kg/h, see Fig. A1) displayed in Fig. 1 represent a best-case detection limit for MethaneSAT, facilitated for the low winds across that scene. We have further explored the drivers of plume detection limits for MethaneSAT using the $Q_{\min}$ formalism described in Sect. 2.2. Figure 3 shows rasters (in sensor coordinates) of XCH$_4$, surface albedo at 1606 nm multiplied by the cosine of the sun zenith angles (as a proxy for near-infrared at-sensor radiance), XCH$_4$ noise, and the resulting $Q_{\min}$ from two MethaneSAT acquisitions: the 5-September-2024

Appalachian acquisition in Fig. 1, and one scene over the Delaware sub-basin of the Permian Basin, acquired on 14 June 2024.

The standard deviation of XCH$_4$ in MethaneSAT L2 maps, $\sigma(\text{XCH}_4)$, which we take as the retrieval precision, typically ranges between 20 and 30 parts-per-billion (ppb). This includes the per-pixel retrieval precision error and the effect of surface clutter on the retrieval. The smaller $\sigma(\text{XCH}_4)$ is found for bright and spatially homogeneous surfaces, which enable the maximum measurement signal-to-noise ratio and the minimum surface clutter. This is illustrated by the $\sigma(\text{XCH}_4)$ maps in Figure 3,

which show a higher retrieval precision for the Permian scene than for the darker Appalachian scene, which is covered by vegetated surfaces with a low near-infrared albedo. Also, we observe a low occurrence of false XCH$_4$ enhancements caused by



surface structures, as opposed to the case of high spatial resolution missions with a coarser spectral resolution, for which it is often challenging to disentangle the methane signal from that of surface features (Roger et al., 2024a).

On the other hand, we find a substantial impact of the wind speed on our ability to detect methane plumes with MethaneSAT.
For stronger winds, plumes and local methane enhancements are more effectively dispersed by the atmosphere, and the number of pixels with $XCH_4$ enhancements above the noise level becomes smaller. Thus, the $Q_{min}$ calculated with Eq. 1 is smaller for the Appalachian acquisition (wind speed is 1.0±0.4 m/s across the scene), than for the Permian Basin one (6.5±1.0 m/s), despite the higher retrieval precision found for the Permian Basin scene. This wind-speed dependence does actually happen for all instruments capable of plume detection (e.g. GHGSat or EnMAP) (Conrad et al., 2023; Ayasse et al., 2024; Roger et al.,
2025), but it may be stronger for MethaneSAT because of its coarser spatial resolution, which reduces the number of plume pixels with large $XCH_4$ values. The fact that MethaneSAT's plume detection ability is greater for a vegetated site (Appalachian) than for a semi-arid surface (Permian) in this comparison is mediated by the lower wind speed in the Appalachian acquisition. This role of wind speed as a limiting factor for plume detection is a characteristic of MethaneSAT observations. In contrast, the surface type is usually the main driver of plume detection for high-resolution instruments.

A preliminary assessment of MethaneSAT's absolute detection limits has been carried out by embedding WRF-LES plumes into real $XCH_4$ maps. Figure 4 shows the result of adding simulated plumes into a subset $XCH_4$ map from a MethaneSAT acquisition over Hassi Messaoud (Algeria) on 17 November 2024. The subset covers an area of about 50 km×50 km, with wind speed of 1.0±0.2 m/s and a 1-sigma retrieval error of 25 ppb (estimated as the local standard deviation from the $XCH_4$ map). This subset included 3 real plumes according to the corresponding MethaneSAT L4-plume product, with emission rates
between 1.1 and 2.7 t/h (Figure 4a). The simulated plumes are marked by a red cross in Figs. 4b-f. The simulations were performed for different wind speeds (either 1.6 or 3.3 m/s) and flux rates (500, 1500 or 2000 kg/h). From visual inspection of the simulated plumes (Figs. 4b-f), it is evident that a plume with flux rate around 500 kg/h would be detectable for this particular scene if the wind speed remained below 1.6 m/s (Figs. 4b), but this may not be the case for the same flux rate range if the wind speed is greater than 3 m/s (Figs. 4d-f), as less plume pixels show $XCH_4$ values above the noise level. We would
like to note that these results are only shown here for illustration purposes. A more rigorous analysis of detection limits would require a probability-of-detection framework (Conrad et al., 2023) and a greater number and range of simulated source rates, wind speeds, plume morphologies and site characteristics.

The impact of wind speed in the MethaneSAT $XCH_4$ map is further illustrated in Fig. 5. It displays two MethaneSAT $XCH_4$ datasets acquired within about 3-weeks over the same Permian-Delaware area, in a region where large concentrations of high-
emitting point sources are usually found (Irakulis-Loitxate et al., 2021; Cusworth et al., 2022; Guanter et al., 2025). The visual comparison of the two maps reveals the impact of wind on the spatial distribution of methane enhancements: a number of methane plumes and local enhancements are found on 22 May 2024, where low-to-medium wind speeds were present (wind speed of 3.0±1.5 m/s), whereas a much more uniform methane concentration field with lower mean $XCH_4$ values is found in the 14 July 2024 acquisition, with much stronger winds (6.9±0.6 m/s). This led to a total of 10 plumes being detected across
the area on the first date, with emission rates between 0.7 and 3.5 t/h, whereas only two large plumes of 4 and 24 t/h were found on the second date, as shown in Fig. 5c. These results are consistent with the analysis in Guanter et al. (2025), which found





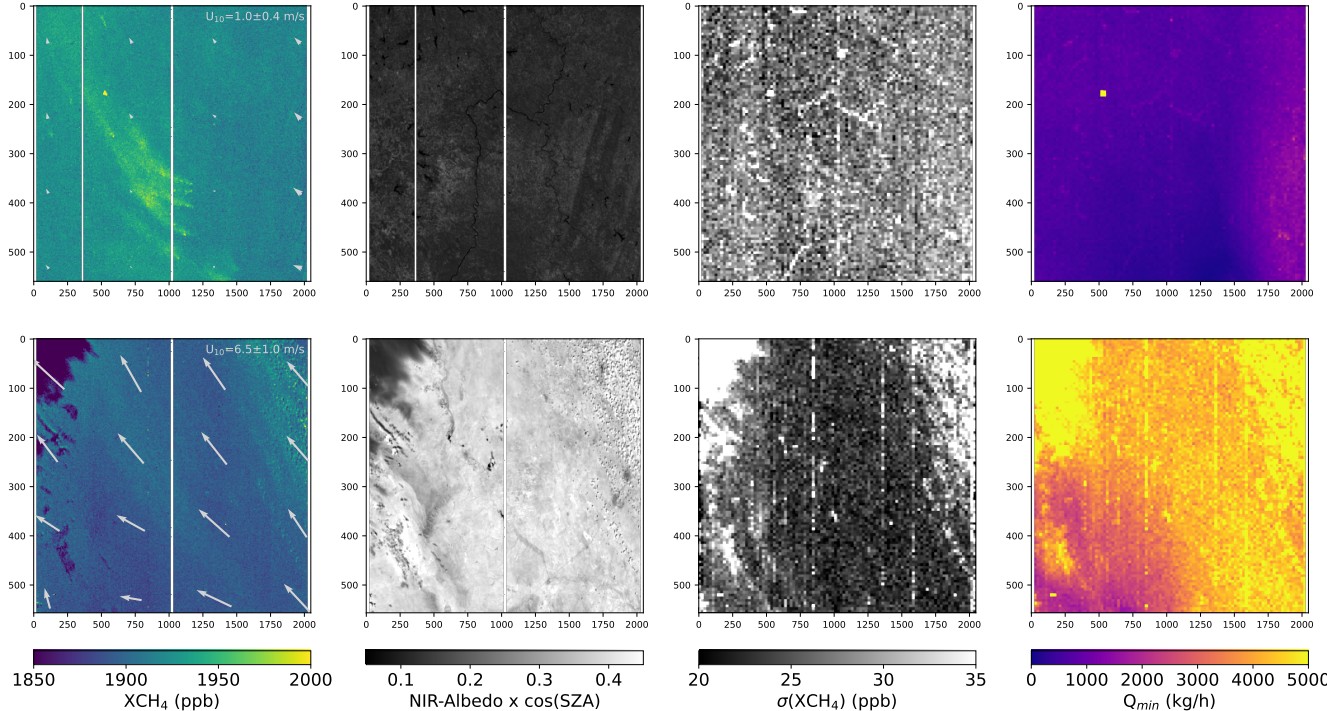

**Figure 3.** Rasters of $Q_{min}$ (Eq. 1) generated from MethaneSAT's L2 datasets acquired over the Appalachian basin on 5 September 2024 and the Permian Basin on 14 June 2024 (top and bottom row, respectively). Methane concentration ($XCH_4$), near-infrared (NIR) albedo (1606 nm) times the cosine of the sun zenith angle (SZA), and $XCH_4$ noise (estimated as the standard deviation of $XCH_4$ within a box of about 3 km) are represented in the first three columns, and the resulting $Q_{min}$ rasters are shown in the last column.

hundreds of plumes over this area from two flight campaigns with MethaneAIR (MethaneSAT's airborne version), but only one source was estimated to be above 2 t/h.

### 3.1.3 Impact of spatial sampling on plume detection and attribution

MethaneSAT has a 110 m×400 m spatial sampling at nadir. This spatial sampling is substantially coarser than that of the high-spatial resolution missions typically used for point source detection, which have squared pixels with a spatial sampling between 20 and 60 m. This may challenge MethaneSAT's ability to pinpoint single sources with respect to those higher resolution missions.

We have explored this issue using again WRF-LES plumes. A simulated plume from a 2000 kg/h source and a 30-m spatial 215 sampling has been convolved with MethaneSAT's 110 m×400 m sampling assuming a boxcar point spread function. This spatial convolution was made so that the plume was aligned with either the across-track or the along-track direction of the observation. The results are represented in Fig. 6. Pixels with $\Delta XCH_4$ values below 25 ppb have been filtered out. The simulated







**Figure 4.** Impact of source intensity and wind speed on plume detection with MethaneSAT. Simulated methane plumes are embedded into a real MethaneSAT L2 dataset from Hassi Messaoud (Algeria) on 17 November 2024. The original subset contains three real methane plumes, which are marked by white annotations in panel (a), whereas the simulated plumes are marked by a red cross in panels (b)-(f).

plume and a zoom on the source location, both at a 30-m sampling, are shown in Figs. 6a-b, and the spatially-resampled plume is displayed in Fig. 6c-d for the plume aligned with the across- and along-track direction, respectively. From these simulations,

it can be observed that the direction of the plume with respect to the observation is important for the potential detection and attribution of the plume, because of MethaneSAT's rectangular pixels: if the plume is aligned with the along-track direction (Fig. 6c), the pixels contain a smaller fraction of the methane enhancement, which decreases the chances of those pixels to stand out above the noise level to enable detection. In the case of plumes aligned with the across-track direction (Fig. 6d), peak $XCH_4$ values are greater than in the previous case because a larger fraction of the enhancement is sampled by each pixel, which

increases the probability of plume detection with respect to the along-track alignment case.




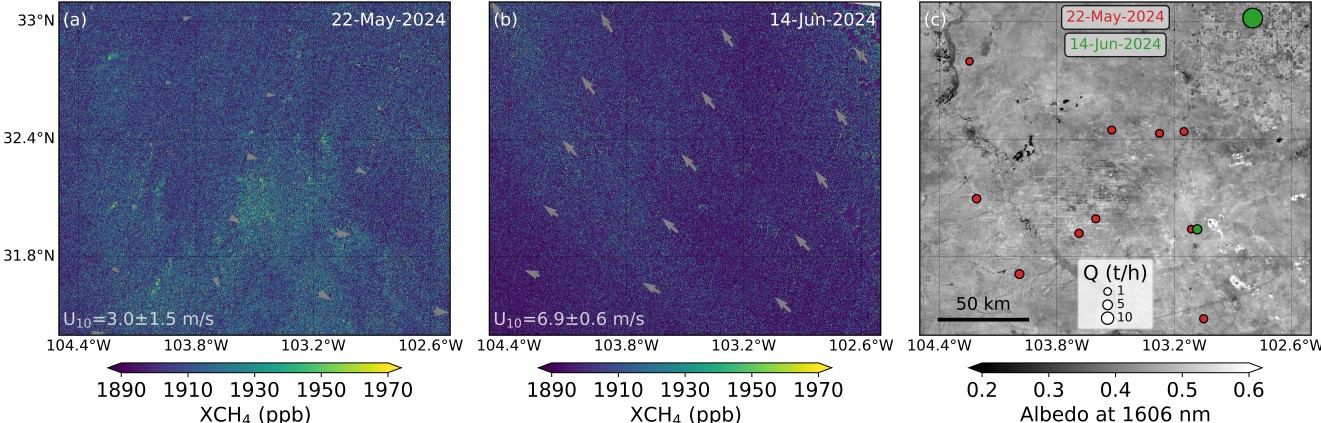

**Figure 5.** Comparison of MethaneSAT XCH$_4$ maps over the Permian-Delaware sub-basin on (a) 22 May and (b) 16 June 2024. The methane plumes detected from each of the datasets are represented by circles on the 22 May albedo map, which is shown in (c). The grey arrows depict the wind speed and direction. The mean wind speed for each subset is provided on the lower-left corner of the (a) and (b) panels.

Regarding the attribution of the plume to a source, the limitations imposed by MethaneSAT's relatively coarse spatial resolution are also illustrated by the simulations in Fig. 6. Whereas the attribution to the source is straightforward for the 30-m sampling case, this becomes challenging at MethaneSAT's spatial sampling, especially in the case of the along-track alignment, in which the first upwind pixel above the noise level is shifted with respect to the source location. This effect is smaller in the case of across-track plume alignment, but even in this case the pixel with the largest concentration enhancement is not the one containing the source, which could lead to ambiguities in the plume attribution process. Again, this example is provided here only to illustrate the problem. More simulations and a robust statistical analysis would be needed for a proper assessment of the limitations of MethaneSAT observations for plume attribution.

### 3.1.4 Comparison with plume data from high-resolution missions

In order to further investigate the limitations interposed by MethaneSAT's relatively coarse spatial sampling, we have directly compared MethaneSAT plume maps with the equivalent data as provided by high-resolution imagers. We have chosen a MethaneSAT acquisition over Turkmenistan's South Caspian basin on 23 September 2024, which was also sampled by the Landsat-8 mission (30-m spatial sampling) on the same day. MethaneSAT overpass was at 10:42 UTC and Landsat's at 7:01, so there was a time difference of about 3.6 h between the two acquisitions. Wind speed for this subset region during the MethaneSAT overpass was 1.3±0.7 m/s. The coordinates and flux rate of the Landsat plumes were taken from the IMEO plume dataset (United Nations Environment Programme, 2025). The results from this comparison are shown in Fig. 7 and Fig. 8.

Figure 7 shows a spatial subset of a MethaneSAT XCH$_4$ map over the region, and the locations of the plumes from both MethaneSAT's L4-plume product and the IMEO plume dataset. We find an overall good agreement in the source locations by





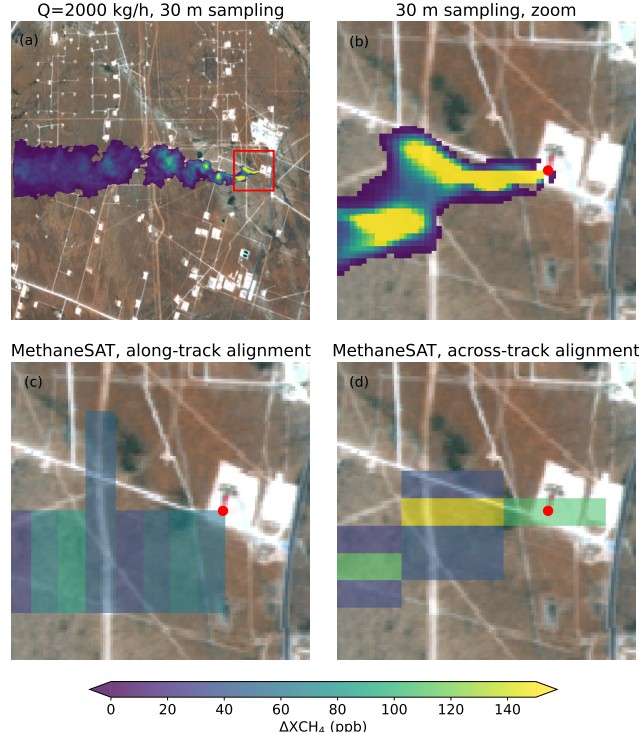

**Figure 6.** Impact of observation geometry on plume attribution over the Permian Basin. A simulated plume with a 30-m spatial sampling (panels a and b) is downscaled to MethaneSAT's 110 m × 400 m sampling assuming that the plume is aligned with either the along- or the across-track direction of the observation (c and d panels, respectively). Pixels with $\Delta$XCH$_4$ values below 25 ppb have been filtered out. The red dot in panels (b), (c) and (d) represents the source of the simulated plume.

the two datasets, although we also observe that two of the plumes detected by MethaneSAT correspond to groups of several

plumes in the finer-resolution IMEO dataset (e.g. the cluster of Landsat plume detections around 38.5°N, 54.2°E, see Fig. 7). This indicates that the MethaneSAT L4-plume product is merging several sources into one single plume in this dataset because of the relatively coarse spatial sampling. The low wind conditions in this dataset may exacerbate this issue, as they cause the emitted gas to accumulate and linger near the source rather than dispersing downwind, complicating the separation of single sources.

A comparison between the emission rates provided by each dataset is shown Fig. 8. The point with a yellow filling depicts the MethaneSAT plume for which several sources in the IMEO dataset have been merged into one, as discussed before. We find a relatively high linear correlation between the two datasets despite the difference in the observation time. The consistency of emission rate estimates from Landsat was assessed by Sherwin et al. (2023) using controlled releases. Considering this, the good correspondence between the MethaneSAT and Landsat plume datasets shown in Fig. 8 represents a first proof of the

consistency of MethaneSAT's L4 plume quantifications.





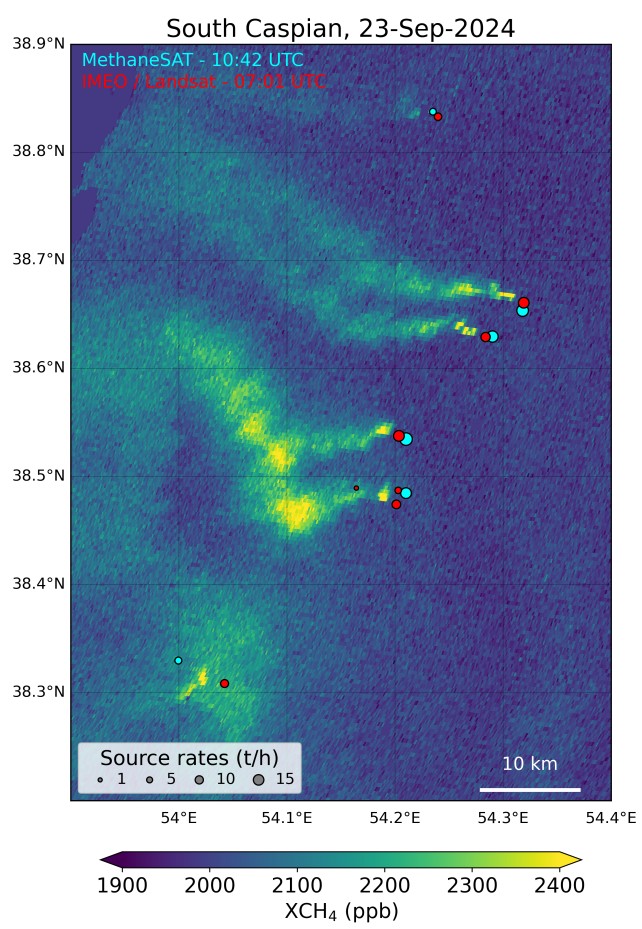

**Figure 7.** Comparison between MethaneSAT and Landsat plume detections over the southern part of Turkmenistan's South Caspian basin on 23 September 2024. The background $XCH_4$ map is a MethaneSAT L2 $XCH_4$ product. The circles depict the plumes included in the corresponding MethaneSAT L4 plume product (cyan) and the IMEO plume database (red).

### 3.1.5 Potential of $\Delta XCH_4$ maps from the matched-filter retrieval for improved plume detection

Comparisons between the matched-filter and the $CO_2$-proxy retrievals are shown in Fig. 9 for two selected cases: a desert area in Hassi Messaoud (Algeria) with a number of relatively weak sources (see also Fig. 4), and a Permian Basin acquisition affected by scattered clouds and strong winds (see also Fig. 5). The $XCH_4$ raster maps from the $CO_2$-proxy retrieval are presented in original and quality-masked formats (left and center column, respectively). In the second case, the quality mask included in MethaneSAT's L2 products has been applied to screen out low-quality pixels (including cloud-contaminated pixels) from the raw $XCH_4$ maps. Relative $\Delta XCH_4$ maps from the raw matched-filter output (no quality mask applied) are shown on the right column for the two areas. The visual comparison of the results from the two retrievals confirms that the matched-filter



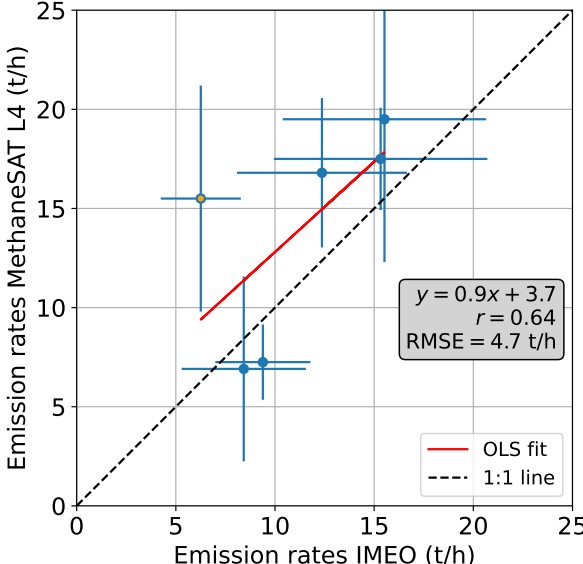

**Figure 8.** Comparison between the source rate estimates in MethaneSAT's L4 plume product and those in the IMEO database for the South Caspian acquisition subset shown in Fig. 7.

retrieval does offer an added-value for the detection of methane plumes, because of its enhanced signal-to-noise ratio and its lower sensitivity to clouds and artifacts. The lower noise in the $\Delta XCH_4$ maps from the matched-filter is visible in the Hassi Messaoud maps. The Hassi Messaoud case also shows that the raw $XCH_4$ map from the $CO_2$-proxy retrieval presents some artifacts resembling methane plumes (Fig. 9a) that are removed by the quality mask (Fig. 9b), whereas these do not appear in the raw $\Delta XCH_4$ map from the matched-filter (Fig. 9c). With respect to clouds and cloud shadows, we observe that the matched-filter retrieval is more robust against cloud contamination, leading to the detection of two plumes in the bottom-right quarter of panel Fig. 9f which would not be detectable in the L2 product, with or without the quality mask (Figs. 9d-e).

On the other hand, we have found the matched-filter retrieval to have strong variations in the absolute $\Delta XCH_4$ values, ranging from a 15% underestimation with respect to the $CO_2$-proxy retrieval for some pristine acquisitions (like the Hassi Messaoud dataset used in this test) up to a 2x difference in others with less favorable acquisition conditions (e.g. regions with vegetated surfaces and high cloud contamination). For this reason, we opt for using arbitrary units with the matched-filter $\Delta XCH_4$ maps in Fig. 9. The matched-filter $\Delta XCH_4$ maps in general very powerful for plume detection, but we do not use them to produce quantitative estimates of emission rates. Further work is required to understand this low-bias behavior of the matched-filter with MethaneSAT, which was not observed for MethaneAIR as reported by Guanter et al. (2025). The relatively low number of along-track samples available in MethaneSAT to constrain the column-wise matched-filter retrieval (about 500 pixels) could partly explain this low bias (Ayasse et al., 2023).







**Figure 9.** Comparison of XCH$_4$ subset maps derived with the MethaneSAT L2 CO$_2$-proxy retrieval with XCH$_4$ enhancement maps from the matched-filter retrieval. The raw and quality-masked versions of the L2 product are displayed on the first two columns, whereas the raw matched-filter output (in arbitrary units) is shown on the third column. The maps in the top row correspond to an acquisition over the Hassi Messaoud basin (Algeria) on 17 November 2024, and the ones in the bottom row are from a 14-June-2024 acquisition over the Permian-Delaware sub-basin (see Figs. 3 and 5).

## 3.2 Assessment of super-emissions across major methane hotspot regions around the world

### 3.2.1 Super-emissions from Turkmenistan's South Caspian basin

Turkmenistan's South Caspian basin is considered the largest hotspot of O&G emissions in the world (Lauvaux et al., 2022). MethaneSAT's XCH$_4$ maps provide an unprecedented view of methane super-emissions in this basin. This is illustrated in Fig.10, which displays a time series of XCH$_4$ maps over the region between September 2024 and May 2025. Red crosses mark the sources that were detected in the 23 September 2024 acquisition. They are plotted in all 6 panels to illustrate the persistency





of the sources in the region within this time period, for which the 23 September acquisition is taken as a reference. A greater number of sources are found in the southern part of the South Caspian basin, although these sources are in general weaker (typically ranging between 5 and 25 t/h) than the two detected in the north, which are estimated to be between 25 and 30 t/h on 23 September, and between 35 and 40 t/h on 18 November (see Fig. A1). However, we acknowledge that some of these plumes may combine emissions from several sources, as also discussed for the same 23 September acquisition in Sect. 3.1.4.

Regarding the persistency of the sources, we observe a large degree of consistency in the sources that are active between September and November 2024 despite the variability introduced by changes in wind speed and direction. The location and number of active sources on 23 September seem to match perfectly the plumes observed on 18 November. This would confirm that the emissions in the South Caspian basin are not only very strong, but also very stable over time. A careful analysis of the different maps in this time series confirms that, in most of the cases, the same sources (and not combinations of neighboring sources) are persistently responsible for all the plumes observed in the datasets between September and November. On the other hand, we do observe some changes in the location of the sources located in the northernmost side of the region in the May 2025 acquisition, with the stronger sources being shifted westwards with respect to the September acquisition, which would indicate an evolution in the O&G extraction activity in the region between November 2024 and May 2025.

### 3.2.2 Super-emissions from the US' Permian Midland sub-basin

The Permian Basin, located between New Mexico and Texas, in the USA, is another major O&G methane super-emission hotspot. Even though O&G operations in the Permian Basin are much more regulated than in Turkmenistan, methane super-emissions in the Permian Basin are also widespread because of the high production activity in the region.

The Permian Basin is comprised of two main sub-basins, namely the Delaware, in the western portion, and the Midland, in the eastern portion. High-resolution airborne surveys and inversion data from TROPOMI show spatial emission hotspots in both sub-basins, although the Delaware sub-basin is the one for which higher levels of methane emissions are expected according to previous reports based on airborne and satellite data (Zhang et al., 2020a; Cusworth et al., 2021b; Varon et al., 2023; Guanter et al., 2025). MethaneSAT $XCH_4$ maps for the Delaware sub-basin have been discussed earlier in this study, including a medium-wind acquisition over this sub-basin from which up to 10 plumes with emission rates between 700 and 3500 kg/h where detected (see Fig. 5a and Fig. A1).

However, our analysis of MethaneSAT data over both sub-basins shows that the Midland sub-basin presents an outstanding concentration of high-emitting point sources in September and October 2024. This is illustrated in Fig. 11, which presents $XCH_4$ maps over an area of about $100 \times 100 \, \text{km}^2$ sampling the Midland sub-basin on four different days (two pairs of consecutive days). A high number of strong point sources is found in three of the four acquisitions. For example, 14 plumes with emission rates between 0.5 and 15 t/h (5 of which have an emission rate greater than 5 t/h) can be found in the 28 September 2024 map (Fig. 11a). These numbers are substantially greater than what has been found for the Delaware sub-basin using MethaneSAT data, and they also exceed those in other studies of super-emitters in the Delaware sub-basin. This includes the one based on MethaneAIR by Guanter et al. (2025), which detected hundreds of plumes in the Delware sub-basin in 2021 and 2023, but only one exceeding 2000 t/h, and the one by Irakulis-Loitxate et al. (2021) using space-based imaging spectrometers,







**Figure 10.** $XCH_4$ enhancement maps ($\Delta XCH_4$) from MethaneSAT L2 products over the South Caspian basin for five different dates between September 2024 and May 2025. The mean and standard deviation of the wind speed ($U_{10}$) from each scene is provided at the bottom-left part of each panel. The gray arrows indicate the wind speed and direction. A key for the estimation of wind speed from the arrow size is provided in panel (c). The red crosses mark the origin of the plumes detected from the 23-September-2024 dataset, and are represented in all panels as a reference for active sources within this region and time period.

which detected 37 plumes across the Delaware sub-basin combining several dates in 2020, but only 3 of them had emission rates greater than 5 t/h. Also a very small fraction of plumes above 5 t/h are reported in the airborne survey by Cusworth et al. (2021a) covering both the Delaware and Midland sub-basins.

The dynamics of these super-emissions in the Midland between September and October 2024 are also intriguing: whereas the sources appear to be very stable between the two consecutive acquisitions on 25 and 26 October, the opposite happens for those



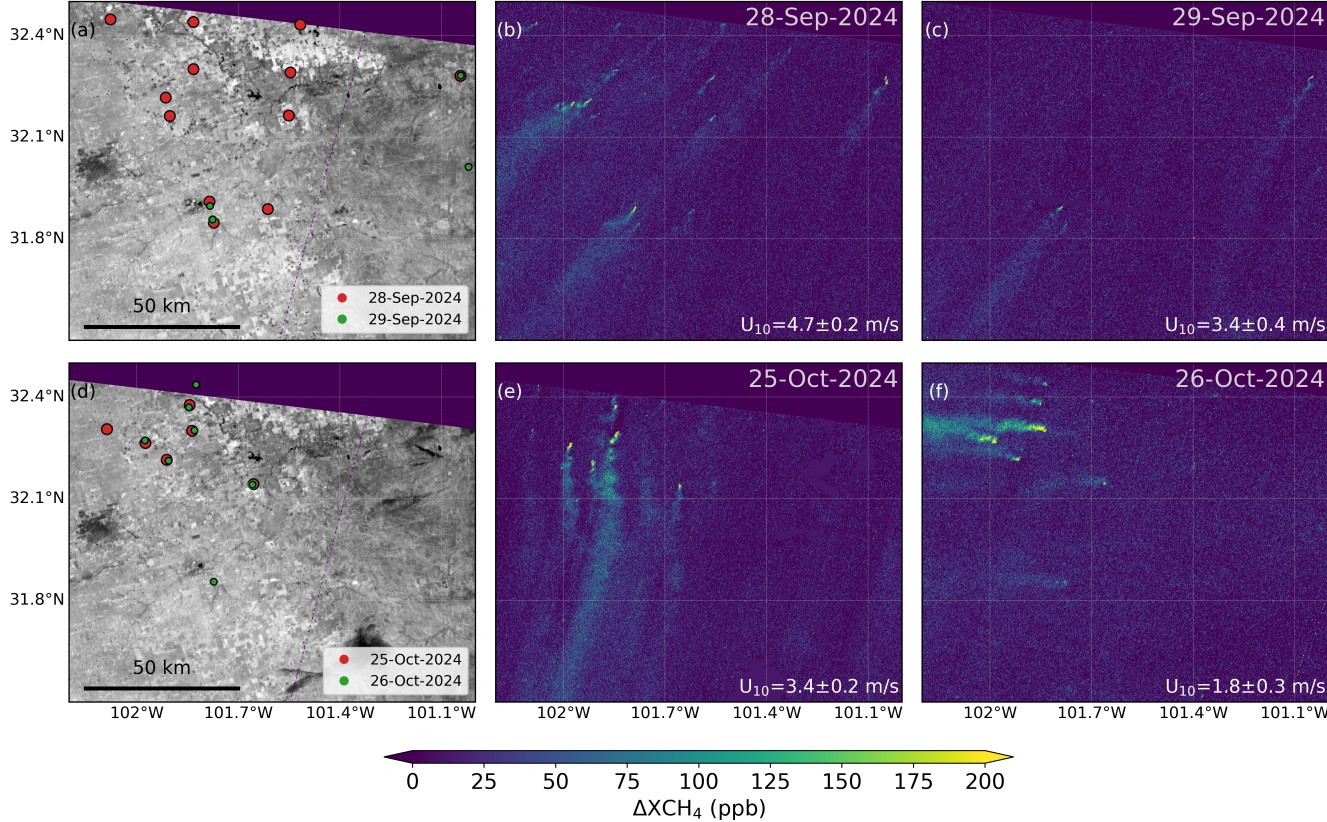

**Figure 11.** XCH$_4$ enhancement maps ($\Delta$XCH$_4$) from MethaneSAT L2 products over the Permian-Midland sub-basin for two pairs of consecutive days. The mean and standard deviation of the wind speed (U$_{10}$) from each scene is provided at the bottom-right part of each panel. The plumes detected from each pair of acquisitions are depicted over the albedo maps represented on the left column.

in September, namely the 14 sources detected on 28 September are reduced to only fout in the next day. This variability can not be related to an increase in the plume detection limits associated to wind speed, as this is actually lower on 29 September than in the day before, so the abrupt change in emissions may be explained by the particular operations taking place in the area between those two days.

Prior research has shown variation in Permian emissions connected with takeaway capacity and concurrent variation in oil 330 prices (Lyon et al., 2021). Before additional pipeline capacity came online in November 2024, west Texas natural gas spot prices were frequently negative throughout the summer and early fall of 2024, creating an environment where Permian operators lost money to transport natural gas (U.S. Energy Information Administration, 2025). We speculate that this operating environment, combined with the relatively high intermittency observed in the Permian overall (Cusworth et al., 2021b) were drivers for the magnitude and fluctuations in day to day high-emitting point source emissions between these Midland observations. Continued





persistence of the underlying sources, albeit at emission rates below the scenes theoretical detection limit, is also a possibility that complicates interpretation of day-to-day emissions.

### 3.2.3 Global view of O&G methane super-emitters based on MethaneSAT

We combined the information on methane enhancements included in the IMEO plume database with our own analysis of the MethaneSAT data archive in order to select the O&G regions in the world with a persistent presence of super-emissions

(Fig. 2). Sample $XCH_4$ enhancement maps from the 6 regions where we have found the strongest presence of high-emitting point sources in the world are shown in Fig. 12. The 6 panels show $XCH_4$ enhancements represented with the same colorbar and cover the same area ($1.5° \times 1.5°$), which enables the visual comparison of the different maps.

The South Caspian and Permian Midland maps were already presented in Figs. 10 and 11, and are reproduced in Fig. 12 (panels a and b) as a reference for comparison with the other basins. A map over the Permian Delaware sub-basin showing

a very large plume is represented (Fig. 12c) to cover the three areas in the world where the highest concentration of super-emissions can be expected. In addition, maps over three other regions are included in the bottom row of Fig. 12: the Maturin basin in Venezuela and the Zagros-Foldbelt and Widyan basins in Iran. Although these three basins are also depicted by the IMEO database as regions with strong methane emission activity (Fig. 2), the regional-level snapshot of $XCH_4$ enhancements provided by MethaneSAT suggests that the super-emissions from those regions are outstanding, and may be comparable to

those of Turkmenistan's South Caspian and the Permian Basin. Five sources with emission rates higher than 5 t/h are found to be simultaneously active across the Maturin basin (see Fig. A1, and Warren et al., in preparation), whereas three sources with emission rates greater than 20 t/h are found in the Zagros-Foldbelt basin (Fig. A1). Several very strong point sources are also identified in the Widyan scene. The constant presence of strong super-emissions in these six regions is confirmed by at least three other MethaneSAT acquisitions over each of the basins (data not shown).

Finally, selected methane concentration maps derived from MethaneSAT observations over other sites included in our O&G site selection (Fig. 2) and from other datasets for which interesting methane emission events have been found, are presented in Fig. 13. In particular, Fig. 13a shows a number of relatively small plumes (range between 500 and 2500 kg/h, see Fig. A1) across the Haynesville-Bossier basin. This range of emissions fits the emission rates typically detectable with public point-source imagers (e.g. EnMAP, EMIT, and Sentinel-2). The acquisition conditions for this Haynesville-Bossier observation are

similar to the ones over the Appalachian region shown in Fig. 1: very low winds ($0.5 \pm 0.2$ m/s) lead to low detection limits, despite the fact that the high fraction of vegetated cover makes this region to be less optimal for plume detection than arid and desert sites. Regarding other acquisition conditions, Fig. 13b shows plumes from three active sources between 1300 and 4000 kg/h (see Fig. A1) within about 30 km in the Amu Darya basin (Eastern Turkmenistan), which was also identified as a global hotspot with TROPOMI (see Fig. 2). Figure 13c shows two plumes from concurrent sources, in this case with very high

emission rates (greater than 100 t/h) found at high latitudes, in West Siberia. No plumes from these sources were found from an overpass of the VIIRS instrument earlier on that day using the methods by de Jong et al. (2025), which suggests that this could have been a short-lived event. We speculate that those two concurrent, large and possibly transient plumes may be caused by pipeline maintenance works in the area. Another very strong emission from an O&G facility is shown in Fig. 13d. It corresponds



**Figure 12.** XCH$_4$ enhancement maps ($\Delta$XCH$_4$) from MethaneSAT L2 products over regions where a high concentration of super-emitters are persistently active. The mean and standard deviation of the wind speed (U$_{10}$) from each scene is provided at the top-right part of each panel. All maps are represented with the same color scale and span approximately $150 \times 150 \, \text{km}^2$ to enable direct visual comparison.

to an O&G extraction platform near the coast of Campeche, in the Mexican side of the Gulf of Mexico. Interestingly, this plume

is not from the nearby Zaap-C platform, from which persistent and very large emissions (sometimes greater than 100 t/h) have been detected in the past (Irakulis-Loitxate et al., 2022a) and was thought to be the only strong source in the area. A different MethaneSAT acquisition does show a plume from the Zaap-C platform (data not shown). The last two panels in Fig. 13 show examples of emissions from the waste sector: a large plume from a landfill in Saudi Arabia, close to Bahrain, is represented in Fig. 13e. An emission rate of 15.2 t/h was estimated for this source, which is consistent with emission estimates for this landfill

from high-resolution missions. Lastly, Fig. 13f shows a massive methane enhancement around Dhaka city, in Bangladesh, that we attribute to a combination of several landfills and dump sites across the city. Similar very large methane enhancements







**Figure 13.** $\Delta XCH_4$ maps from MethaneSAT showing plumes from a range of regions, sources and acquisition conditions.

(which are probably among the largest enhancements found in the entire MethaneSAT archive) have been found over the same site in other MethaneSAT observations. This site was also listed as one of the largest and most persistent methane emitting regions in the world by Vanselow et al. (2024).

## 4  Conclusions

In this study we have provided a first assessment of MethaneSAT's capacity to monitor methane super-emitters from O&G basins around the world. We have first characterized the potential and limitations of MethaneSAT for the detection and quantification of methane plumes, and have then exploited the existing MethaneSAT data archive to analyze the super-emissions (intensity and persistency of high-emitting point sources) from key O&G emission areas around the world.



Our results confirm that MethaneSAT is well suited to fill the observational gap between TROPOMI and the group of high-spatial resolution missions which are being widely-used for the detection and quantification of single plumes from point sources. This is facilitated by MethaneSAT's wide area coverage, medium spatial resolution and high sensitivity to methane, which result into accurate and precise $XCH_4$ retrievals and a low sensitivity to the surface background, enabling robust plume detection over vegetated and urban surfaces. Thanks to these characteristics, MethaneSAT can provide a new view on re-
gional emissions, better accounting for diffuse emissions from distributed infrastructure like pipelines, compressor stations, or wetlands, bridging the gap between fine-scale detection (like GHGSat), and global-scale monitoring (like TROPOMI).

 On the other hand, we also observe some limitations of MethaneSAT for point-source analysis. These are mostly interposed by MethaneSAT's coarser spatial sampling as compared with point source imagers, which may challenge the separation of individual plumes inside larger methane enhancements and the attribution of plumes to single sources. We have also found
that wind speed plays a more important role in the detection and interpretation of methane plumes with MethaneSAT than in the case of higher resolution missions. These findings suggest that a synergistic use of MethaneSAT with available point-source datasets (e.g. the IMEO plume dataset being developed by the IMEO-MARS program of the UNEP) is needed for a full characterization of the point sources across a given O&G basin.

 From our analysis of the MethaneSAT data archive, we have gained new insights into global O&G methane emission
hotspots. We have evidenced the high number and persistency of very strong sources in Turkmenistan's South Caspian Basin, and have revealed the high concentration of super-emitters within the Permian Basin's Midland sub-basin, which appears to be a new development in the region, as emissions in the Permian Basin were traditionally dominated by those from the Delaware sub-basin. Also, we have identified three other basins, namely Maturin in Venezuela and Zagros-Foldbelt and Widyan in Iran, where very strong point sources have been detected along a series of MethaneSAT observations. These results provide the basis
for a number of potential follow-up studies focused on those regions.

 Despite the important information on methane high-emitting point sources that MethaneSAT is able to convey, we acknow-ledge that super-emissions only represent a fraction of the total emissions from O&G basins (Omara, 2022; Williams et al., 2025). MethaneSAT's high-accuracy $XCH_4$ retrievals also enable the inversion of regional methane fluxes, which is part of the MethaneSAT L4 product. The combination of those basin-level fluxes with the super-emissions discussed in this study is
needed for a full picture of O&G emissions around the planet.

**Appendix A: Emission rates from key datasets analyzed in this study**

*Author contributions.* LG, RG, and MO designed the study. LG led the analysis and produced the figures. JG and MS contributed to the implementation of the matched filter for MethaneSAT. JW and HH contributed to the analysis of point sources in the Permian Basin and Turkmenistan, respectively. MS and ZZ generated emission rate data from the detected plumes, and SS, CCM and SW led the development
of the MethaneSAT L2 products. LG wrote the paper, incorporating comments and revisions from authors.







**Figure A1.** Source rates and $Q_{\min}$ for some of the acquisitions shown in this work. The top panel shows the source rates for individual plumes, and the bottom panels show the total source rate obtained from combining all the single plumes (left) and the average $Q_{\min}$ from the areas shown in the $XCH_4$ maps presented in previous figures. The gray bar in the bottom left panel marks the total emission for the Appalachian scene if the massive, short-lived emission is counted (see Fig. 1).

*Competing interests.* We declare no competing interests.



*Acknowledgements.* Funding for MethaneSAT and MethaneAIR activities was provided in part by Anonymous, Arnold Ventures, The Audacious Project, Ballmer Group, Bezos Earth Fund, The Children's Investment Fund Foundation, Heising-Simons Family Fund, King Philanthropies, Robertson Foundation, Skyline Foundation and Valhalla Foundation. For a more complete list of funders, please visit

www.methanesat.org.



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
