# Peer review of "Surveying Methane Point-Source Super-Emissions across Oil and Gas Basins with MethaneSAT"

_EGUsphere, 2025_

## Referee Comment (RC1)

**General Comments:**

This paper is the first assessment of MethaneSATs point source methane plume mapping and quantification. The subject of this paper is important to the methane mapping community due to the high-profile nature of the MethaneSAT mission. It is my recommendation that this paper be published with significant revisions. This paper aims to do two things, first it aims to quantify MethaneSATs detection limits using a combination of synthetic and real imagery. Second it aims to provide an assessment of plumes detected by MethaneSAT in key global methane hotspots.

This study provides a good assessment of MethaneSAT's plume detection capabilities. However, it could be strengthened by 1. Using the MethaneSAT data archive to do the analysis. 2. Including a probability of detection analysis. 3. Doing a quantitative plume comparison with a higher quality point source imager. Details on these points are provided below.

The assessment of the global hotspots provides nice examples of MethaneSAT plume data however the analysis of the hotspots lacks scientific rigor and unique insights. Details on suggested improvement are provided below.

**Major/Specific Comments:**

**Abstract**

In line 9 a probability of detection would be preferable to a minimum detection (I have more comments below on POD). Please define favorable condition (windspeed, albedo, etc.)

Line 14 to 16. This statement seems to overstate the results of the paper and the potential of the MethaneSAT archive. Here and throughout the paper, the authors should take caution not to overstate the significance of the plumes highlighted in this study.

**Methods and Materials**

In line 74 this paper would benefit from a summary of the methods in addition to a reference to a paper.

For the data used in this study, one goal of this paper is to evaluate plume detection with regards to surface albedo, retrieval noise, pixel size, and wind speed. The paper would be improved if the authors showed that the data used represents a diversity of albedo, XCH4 noise, and windspeed (for example a histogram of each variable for all data used in the study). In addition, the authors could go further and use the XCH4 and XCH4 noise to show the precision for all data used in the analysis.

**Assessment of plume detection limits**

The analysis of the drivers of the MethaneSATs detection limit overall is good and I especially like figure 3 as a visual interpretation of the main drivers. However, the analysis relies too heavily on the Jacobs 2016 equation (equation 1 in this paper). Given that the MethaneSAT archive is complete, this section would be strengthened by an empirical assessment of the drivers of detection limits. For example, the albedo, XCH4 noise, and wind speed can be calculated for every detected plume in the MethaneSAT archive and the authors could show or deduce a relationship between the afore mentioned factors and the emission rate. Results from the actual data would make the analysis much stronger. Relying too heavily on a model does not highlight the actual performance of the data which is a major aim of this paper.

Probability of detection (POD) has emerged as a more common and more accurate measure of detection limit. Point source images generally report a 90% or 50% POD at a 3 m/s wind speed. This paper would be greatly improved if the authors attempt to construct a POD curve for MethaneSAT. This would not only provide a state of the science analysis but also make it more easily comparable to other point source imagers. The authors could use the complete MethaneSAT archive and MethaneAir or other IMEO data to construct a figure similar to figure 2 in Kunkel et al 2023 or figure 2 in Ayasse et al 2024. This type of analysis would significantly strengthen this paper and provide a key finding to highlight.

Impact of spatial sampling on plume detection and attribution I think the impact of spatial sampling is a good analysis and found those results interesting.

**Comparison with plume data from high-resolution missions**

The intention of this analysis is good, but the quality of the Landsat methane plume data detracts from the quality of this analysis. Comparing MethaneSAT to a higher quality or better validated point source imager (such as MethaneAIR or an imaging spectrometer) would make this analysis not only show consistency but also validate MethaneSATs quantitative emission estimates. Better yet, if controlled release data is available for MethaneSAT this would be an excellent place to display the results. These data may not be available, and if this is the case, an explanation of why this is the best available data for this analysis would strengthen the paper.

**Potential of $\Delta XCH4$ maps from the matched-filter retrieval for improved plume detection**

This section could get added to the methods section. The matched filter is the industry standard detection algorithm, but the issues listed here are enough to justify using the CO2 proxy for qualification. The methods section could be expanded to include this analysis and the justification for the final quantification choice.

**Assessment of super-emissions across major methane hotspot regions around the world**

This section lacks scientific rigor and unique insights. This section should get reduced (although if reduced, I recommend expanding the plume detection limit analysis) or narrowed in scope with a more in-depth analysis. MethaneSAT is advertised as being able to do regional trends and point sources (in fact this paper highlights this capability in the introduction). This section would be very strong if the authors could do a regional flux + point source analysis exclusively with MethaneSAT data for a few key regions. This type of analysis would highlight MethaneSATs unique capabilities and it would be truly novel work. This maybe out of the scope of the paper therefore the comments below offer other suggestions that only utilize point source data.

**Super-emissions from Turkmenistan's South Caspian basin**

Turkmenistan is one of the easiest places on earth to observe methane plumes and there is a rich methane plume data archive from a multitude of instruments for Turkmenistan. The MethaneSAT images alone do not yield any new or interesting insights regarding this region. Both persistence and the evolution of O&G extraction in this region, while interesting topics, could be done better by incorporating more data. I would consider removing this section.

**Super-emissions from the US' Permian Midland sub-basin**

Again, this is one of the most highly studied methane emitting regions globally, therefore the bar for meaningful analysis is higher than just plume detection and quantification. To strength this paper and to highlight the capability of MethaneSAT think it would be better to attempt to do a quantitative analysis of the total basin super emitter emissions or intensity in the basin. This could then be compared to work from previous studies (for example, Zhang 2020, Cusworth 2022, Irakulis-Loitxate 2021, Chen 2022, Varon 2022, or many others). This is one idea but other in depth or unique analysis could be added here to strengthen this section.

In addition, the data presented in this section does not seem strong enough to state the observed plumes are due to the addition of pipeline capacity. This region is known for extreme day to day variability in emissions and I think more images/data are needed to link observed emissions to this specific activity.

**Global view of O&G methane super-emitters based on Methane**

Although not a rigorous scientific study, this section and these figures do highlight the diversity of plumes in the methaneSAT archive. However, this section would be strengthened if the authors could elaborate on how a year of MethaneSAT observations are a valuable contribution to the field and provide a discussion of what MethaneSAT data provides that the current constellation of point source imagers do not.

**Conclusions**

Line 385 states that the results confirm that MethaneSAT can fill the observational gap between TROPOMI and high resolution missions, however it is not clear to me know the analysis or the results show this. Line 390 also states capabilities that are not discussed in this paper. I would add the contents of the sentence in line 397-389 to the paragraph below and remove the remainder of this paragraph.

In Line 399 the paper does not provide "new insights to the global O&G methane hotspots." The authors should be more cautious about overstating the results of this analysis.

**Minor Comments:**

Line 30 -> delete "on the one hand"

Lin3 33 -> define "very large point sources"

Line 36 -> delete "on the other hand"

Line 63 -> what does the .08 nm in parenthesis represent?

Figure 1-> a scale bar on (b) would make it easier to compare to the methaneSAT image.

line 110 -> Delete on the other hand and replace with however

line 117 -> delete "on the other hand"

Line 155 -> curious about this plume, it doesn't look like the other plumes in the XCH4 in figure 1 and given it is not in any other data from the day has more analysis been done to 1. Confirm it is a real plume by attributing it to infrastructure that could potentially have a <70 min 100 t/hr emission and 2. doing a due diligence evaluation to make sure it is not an artifact or false positive

Line 174 -> delete "on the other hand"

Figure 8 -> one point is orange while the others are blue, please explain this in the figure caption or correct if an error.

Line 271 -> delete "on the other hand"

Line 296 -> delete "on the other hand"

Figure 10 and 11 -> These figures could be combined, and the south Caspian and Permian basin images could be dropped as these areas are discussed elsewhere.

Line 392 -> delete "on the other hand"

---

## Author Comment (AC1)

**Response to reviewers (egusphere-2025-4666)**

Please, find enclosed our point-by-point responses to the comments and suggestions made by the three reviewers of our manuscript. The comments from the reviewers are in black, and our responses are in blue.

The main changes made on the manuscript to address the reviewers' comments are:

- **Deeper discussion of retrieval precision and detection limits:**
    - A new section "Assessment of factors driving the detection of methane plumes with MethaneSAT" has been added. A more in-depth description of plume detection drivers is now provided in the section.
    - Retrieval precision: Fig. 3 has been extended with two more scenes in order to better sample different acquisition conditions (surface type and wind intensity) and relate those to retrieval precision and detection limits.
    - Detection limits: a new figure (Fig. 5) has been added in order to infer the "average detection limits" from a range of MethaneSAT acquisitions.

- **Discussion of Permian-Midland results:** The analysis of the Permian-Midland scenes has been extended by adding a new Fig. A2 showing emission rates from single plumes and point source totals from all the Permian scenes (Delaware and Midland) presented in the study. Total regional emissions from MethaneSAT L4-area products have also been added to the discussion.

- **Novelty of the study cases presented in the "Assessment of super-emissions across major methane hotspot regions around the world" section:**
    - We have discussed in the text what the main findings of our scene analysis are, both for basins with a high density of super-emitters (e.g., the South Caspian Basin in Turkmenistan and the Permian Basin in the United States), and those which are comparatively under-represented in the existing literature (e.g., Maturin in Venezuela and Zagros-Foldbelt in Iran).
    - Panels including further XCH4 maps from the latter basins have been added to Fig. 13 (former Fig. 12).

**Reviewer 1**

General Comments:
This paper is the first assessment of MethaneSATs point source methane plume mapping and quantification. The subject of this paper is important to the methane mapping community due to the high-profile nature of the MethaneSAT mission. It is my recommendation that this paper be published with significant revisions. This paper aims to do two things, first it aims to quantify MethaneSATs detection limits using a combination of synthetic and real imagery. Second it aims to provide an assessment of plumes detected by MethaneSAT in key global methane hotspots.

This study provides a good assessment of MethaneSAT's plume detection capabilities. However, it could be strengthened by 1. Using the MethaneSAT data archive to do the analysis. 2. Including a probability of detection analysis. 3. Doing a quantitative plume comparison with a higher quality point source imager. Details on these points are provided below.

The assessment of the global hotspots provides nice examples of MethaneSAT plume data however the analysis of the hotspots lacks scientific rigor and unique insights. Details on suggested improvement are provided below.

Thanks for the careful review of our manuscript.

The overarching goal of this study is to provide an initial assessment of MethaneSAT's potential and limitations for the analysis of methane point sources. This includes a relatively wide range of aspects, from MethaneSAT's ability to detect, quantify and attribute methane plumes, to the illustration of sites and data included in the archive. The broad scope of the manuscript limits the depth with which each aspect can be covered.

Still, we have done our best to improve the manuscript following the comments and suggestions by the reviewer. Please, see below for more details.

Major/Specific Comments:

*Abstract*

In line 9 a probability of detection would be preferable to a minimum detection (I have more comments below on POD). Please define favorable condition (windspeed, albedo, etc.)

We have added "*We estimate a minimum detection limit of about 500 kg/h for a bright surface and low wind conditions, and an average detection limit around 1300 kg/h corresponding to an ensemble of sites with different observation conditions*" (please, see below for comments on POD)

Line 14 to 16. This statement seems to overstate the results of the paper and the potential of the MethaneSAT archive. Here and throughout the paper, the authors should take caution not to overstate the significance of the plumes highlighted in this study.

Rephrased to "*Our results illustrate the potential of MethaneSAT data to map methane emissions from hotspot regions and super-emitters around the planet.*"

*Methods and Materials*

In line 74 this paper would benefit from a summary of the methods in addition to a reference to a paper.

These following paragraphs have been added:

- Plume detection: *"The first version of these methods relied on methane concentration thresholds to detect and segment the plumes, but more sophisticated methods consisting in data denoising and different criteria for plume detection and masking (including concentration, plume shape and wind direction) were implemented later \citep{zhanz_Wavelet}. The plumes detected by this automated processing are quality-controlled by a human before becoming public."*

- Plume quantification: "In particular, the DI-growing box method used for the quantification of MethaneSAT plumes applies Gauss's divergence theorem to estimate methane emissions by integrating methane flux divergence within expanding boxes that enclose the plume, eliminating the need to define inflow concentrations. By averaging methane gradients with wind vectors around the plume, it is well suited for larger or clustered sources but less sensitive to small point emitters. This method has been validated by controlled release experiments \citep{ElAbbadi_2024}."

For the data used in this study, one goal of this paper is to evaluate plume detection with regards to surface albedo, retrieval noise, pixel size, and wind speed. The paper would be improved if the authors showed that the data used represents a diversity of albedo, XCH4 noise, and windspeed (for example a histogram of each variable for all data used in the study). In addition, the authors could go further and use the XCH4 and XCH4 noise to show the precision for all data used in the analysis.

We have extended the discussion of the plume detection drivers. There is now a dedicated section for this topic, "*3.1.2 Assessment of factors driving the detection of methane plumes with MethaneSAT*", and two more datasets have been included in Fig.3 so that it now shows a higher variety of surface types, retrieval precision and wind speed.

These lines describe the motivation behind the selection of the sites included in Fig. 3, in line with the reviewer's comment "the data used represents a diversity of albedo, XCH4

noise, and windspeed"): "*These four acquisitions have been selected in order to sample a wide range of acquisition conditions, in particular of at-sensor radiance (from very low in the Appalachian and West Siberia scenes to very bright in the Permian and Algeria acquisitions) and of wind intensity (from very low in the Appalachian to very strong in the Permian).*"

*Assessment of plume detection limits*

The analysis of the drivers of the MethaneSATs detection limit overall is good and I especially like figure 3 as a visual interpretation of the main drivers. However, the analysis relies too heavily on the Jacobs 2016 equation (equation 1 in this paper). Given that the MethaneSAT archive is complete, this section would be strengthened by an empirical assessment of the drivers of detection limits. For example, the albedo, XCH4 noise, and wind speed can be calculated for every detected plume in the MethaneSAT archive and the authors could show or deduce a relationship between the afore mentioned factors and the emission rate. Results from the actual data would make the analysis much stronger. Relying too heavily on a model does not highlight the actual performance of the data which is a major aim of this paper.

We agree with the reviewer that it would be ideal to support the model-based analysis with real data. However, the list of quality-controlled methane plumes derived from MethaneSAT is not so extensive (about 165 plumes at the moment, see section 2.5) and we do not think that the analysis proposed by the reviewer would lead to robust conclusions with such a small sample size.

On the other hand, the current Fig. 4 (former Fig. 5) does illustrate the effect of wind speed on plume detections in the Permian basin, which we think partly supports the findings from the model-based results in Fig. 3.

Probability of detection (POD) has emerged as a more common and more accurate measure of detection limit. Point source images generally report a 90% or 50% POD at a 3 m/s wind speed. This paper would be greatly improved if the authors attempt to construct a POD curve for MethaneSAT. This would not only provide a state of the science analysis but also make it more easily comparable to other point source imagers. The authors could use the complete MethaneSAT archive and MethaneAir or other IMEO data to construct a figure similar to figure 2 in Kunkel et al 2023 or figure 2 in Ayasse et al 2024. This type of analysis would significantly strengthen this paper and provide a key finding to highlight.

We agree that constructing a POD curve for MethaneSAT would be very interesting. However, a reference dataset containing non-detected plumes (i.e. with emission rates below the instrument's detection limit) is needed to derive a POD. Such reference datasets can be obtained from controlled releases, airborne underflights, or simulations, which unfortunately are not available for MethaneSAT (a study using simulations is ongoing, though).

We have addressed this point by new data analysis, this time including all available plumes in the MethaneSAT archive as suggested by the reviewer. We have fitted a lognormal curve to the histogram of all available plume detections, which is shown in the new Fig. 5. The point at which the curve rolls off toward smaller values (about 1300 kg/h) has been taken as an "average" detection limit for the collection of MethaneSAT datasets contributing to the plume list.

*Impact of spatial sampling on plume detection and attribution*

I think the impact of spatial sampling is a good analysis and found those results interesting.

*Comparison with plume data from high-resolution missions*

The intention of this analysis is good, but the quality of the Landsat methane plume data detracts from the quality of this analysis. Comparing MethaneSAT to a higher quality or better validated point source imager (such as MethaneAIR or an imaging spectrometer) would make this analysis not only show consistency but also validate MethaneSATs quantitative emission estimates. Better yet, if controlled release data is available for MethaneSAT this would be an excellent place to display the results. These data may not be available, and if this is the case, an explanation of why this is the best available data for this analysis would strengthen the paper.

We agree that Landsat data cannot be a reference for the validation of MethaneSAT emission rates due to its relatively low sensitivity to methane. However, we haven't found match-ups of observations from MethaneSAT and other targeted missions, such as EnMAP or EMIT, within a low time difference and over regions showing several strong and persistent active sources. For this reason, we opt for keeping the comparison with Landsat as is in the manuscript. We have justified this in the text with the new paragraphs copied below:

"*We have chosen this Landsat acquisition for comparison with MethaneSAT because it spans a number of plumes within a relatively broad range of emission rates. No match-ups with imaging spectroscopy missions, such as EnMAP or EMIT (more sensitive to methane than Landsat, but with a sparse spatio-temporal sampling), have been found for this comparison with MethaneSAT.*"

"*However, it must be stated that this comparison does not represent a validation of the MethaneSAT emission rates: first, the Landsat emission rates may have substantial errors \citep{sherwin2023single}; second, the emission rates from these super-emissions can vary strongly within the 3.6\,h time difference between the MethaneSAT and Landsat acquisition.*"

*Potential of ΔXCH4 maps from the matched-filter retrieval for improved plume detection*

This section could get added to the methods section. The matched filter is the industry standard detection algorithm, but the issues listed here are enough to justify using the CO2 proxy for qualification. The methods section could be expanded to include this analysis

and the justification for the final quantification choice.

Thanks for this suggestion. However, considering that the results shown in this section involve data processing and analysis, we prefer to keep the matched-filter part split between the Methods and Results sections.

*Assessment of super-emissions across major methane hotspot regions around the world*

This section lacks scientific rigor and unique insights. This section should get reduced (although if reduced, I recommend expanding the plume detection limit analysis) or narrowed in scope with a more in-depth analysis. MethaneSAT is advertised as being able to do regional trends and point sources (in fact this paper highlights this capability in the introduction). This section would be very strong if the authors could do a regional flux + point source analysis exclusively with MethaneSAT data for a few key regions. This type of analysis would highlight MethaneSATs unique capabilities and it would be truly novel work. This maybe out of the scope of the paper therefore the comments below offer other suggestions that only utilize point source data.

Thanks for this comment.

We are not sure what type of "scientific rigor" the reviewer is missing in this section. In any case, we have added a new introductory paragraph for this section (reproduced below) in order to better state the objectives of this part of the study.

"*In this section, we demonstrate the potential of MethaneSAT to map emissions from super-emitting point sources at the regional scale. To this end, we surveyed the full MethaneSAT data archive and selected representative XCH$_4$ maps that illustrate (i) MethaneSAT's ability to provide wide-area characterization of methane plumes over oil- and gas-producing basins with a high density of super-emitters (e.g., the South Caspian Basin in Turkmenistan and the Permian Basin in the United States), and (ii) super-emission events in regions that are comparatively under-represented in the existing literature  (e.g., Maturin in Venezuela and Zagros-Foldbelt in Iran).*"

*Super-emissions from Turkmenistan's South Caspian basin*

Turkmenistan is one of the easiest places on earth to observe methane plumes and there is a rich methane plume data archive from a multitude of instruments for Turkmenistan. The MethaneSAT images alone do not yield any new or interesting insights regarding this region. Both persistence and the evolution of O&G extraction in this region, while

interesting topics, could be done better by incorporating more data. I would consider removing this section.

Thanks for this comment. Even if the point sources in Turkmenistan have already been reported in many papers, we think that it is relevant for our study to illustrate MethaneSAT's view of this key methane hotspot, so we prefer to keep this section in the manuscript.

Following the reviewer's comment, we have simplified the text of this section, and added the statement on adding other data sources to this analysis as reproduced below.

"*However, we acknowledge that some of these plumes may combine emissions from several sources, as also discussed for the same 23 September acquisition in Sect.\,\ref{sec:comp_IMEO}, so it is not clear whether it is always the exact same sources that are active, or neighboring sources within the same area. Adding plume detections from higher-resolution satellite instruments, such as those included in the IMEO plume database \citep{imeo_plumes}, would help better understand these patterns.*"

*Super-emissions from the US' Permian Midland sub-basin*

Again, this is one of the most highly studied methane emitting regions globally, therefore the bar for meaningful analysis is higher than just plume detection and quantification. To strength this paper and to highlight the capability of MethaneSAT think it would be better to attempt to do a quantitative analysis of the total basin super emitter emissions or intensity in the basin. This could then be compared to work from previous studies (for example, Zhang 2020, Cusworth 2022, Irakulis-Loitxate 2021, Chen 2022, Varon 2022, or many others). This is one idea but other in depth or unique analysis could be added here to strengthen this section.

Thanks for this suggestion. We have added a discussion of the variability of point source totals and total regional emissions from the Permian scenes, also addressing a comment from Reviewer 2. For that, we have added a new Fig. A2 showing the total emission rate from the point sources in Fig. 12, and the total regional emission numbers from the L4 products provided in MethaneSAT's data portal https://portal.methanesat.org/.

"*This variability can also be seen in the total point source emissions reported in Fig.\,\ref{fig:Q_bars_Permian}: the maximum total point source emission is found on 28 September (87.8\,t/h), which is reduced by a factor \sim{5x} on the next day (16.3\,t/h). Intermediate and more similar values are found on 25 and 25 October (50.4 and 57.6\,t/h). The strong reduction in emissions from 28 to 29 September are also found in the corresponding total regional fluxes, which are included in the L4 products available in MethaneSAT's Data Portal \citep{msat_data_portal}: the total emissions are 180\,t/h for 28 September, and 140\,t/h for 29 September. It must be remarked that this total emission estimate combines the detected point source emissions with the total area emissions estimated through a full atmospheric model inversion. Interestingly, the total emission rate estimated from the 26 October scene, 192\,t/h, is*

*slightly higher than the one obtained for 28 September, unlike what happens with the point sources. There is no total emission estimate from the 25 October scene, as the L4-total flux product from this date did not pass the quality control.*"

In addition, the data presented in this section does not seem strong enough to state the observed plumes are due to the addition of pipeline capacity. This region is known for extreme day to day variability in emissions and I think more images/data are needed to link observed emissions to this specific activity.

This paragraph has been rewritten as shown below in order to reduce the amount of speculation without data while providing the underlying attributions like the reviewer suggested

 "*Prior research has shown variation in Permian emissions connected with takeaway capacity and concurrent variation in oil prices (Lyon et al., 2021). Before additional pipeline capacity came online in November 2024, west Texas natural gas spot prices were frequently negative throughout the summer and early fall of 2024, creating an environment where Permian operators lost money to transport natural gas (U.S. Energy Information Administration, 2025). We note that 76% (23/30) of the marked plumes (Figure 11) originated from midstream facilities, including compressor stations, processing plants, and gathering pipelines. The heighted rate of plumes from midstream facilities may be indicative of the beforementioned limitations in takeaway capacity for the broader Permian at that time of the year.*"

*Global view of O&G methane super-emitters based on Methane*

Although not a rigorous scientific study, this section and these figures do highlight the diversity of plumes in the methaneSAT archive. However, this section would be strengthened if the authors could elaborate on how a year of MethaneSAT observations are a valuable contribution to the field and provide a discussion of what MethaneSAT data provides that the current constellation of point source imagers do not.

We have added the lines copied below.

"*The $XCH_4$ maps from MethaneSAT shown in the previous figures offer a unique combination of wide spatial coverage, medium resolution and low sensitivity to surface-related artifacts. This complements the data from available point source imagers, which generally have a finer spatial sampling at the expense of a narrower coverage and a higher occurrence of false positive caused by surface structures.*"

"*The very large emissions from Maturin, Zagros-Foldbelt and Widyan have received limited attention in previous studies, but the MethaneSAT $XCH_4$ data indicate that these areas warrant further targeted investigation.*"

*Conclusions*

Line 385 states that the results confirm that MethaneSAT can fill the observational gap between TROPOMI and high resolution missions, however it is not clear to me know the analysis or the results show this. Line 390 also states capabilities that are not discussed in this paper. I would add the contents of the sentence in line 397-389 to the paragraph below and remove the remainder of this paragraph.

In Line 399 the paper does not provide "new insights to the global O&G methane hotspots." The authors should be more cautious about overstating the results of this analysis.

We have rephrased those lines as:

*"Our results illustrate that MethaneSAT's observations are complementary to those from TROPOMI and the group of high-spatial resolution missions which are being widely-used for the detection and quantification of single plumes from point sources."*

and

*"In addition to evaluating MethaneSAT's performance for point source work, we have used the available data archive to further investigate emissions across global O\&G methane emission hotspots."*

*Minor Comments:*

We have implemented the suggestions below in most of the cases. Some specific points are discussed below.

Line 30 -> delete "on the one hand"
Lin3 33 -> define "very large point sources"
Line 36 -> delete "on the other hand"
Line 63 -> what does the .08 nm in parenthesis represent?
Figure 1-> a scale bar on (b) would make it easier to compare to the methaneSAT image.
line 110 -> Delete on the other hand and replace with however
line 117 -> delete "on the other hand"
Line 155 -> curious about this plume, it doesn't look like the other plumes in the XCH4 in figure 1 and given it is not in any other data from the day has more analysis been done to 1. Confirm it is a real plume by attributing it to infrastructure that could potentially have a <70 min 100 t/hr emission and 2. doing a due diligence evaluation to make sure it is not an artifact or false positive
This sentence has been added: "*We ruled out the possibility of this plume being a retrieval artifact by comparing with the near-infrared radiance map, where no similar pattern is found.*"

Line 174 -> delete "on the other hand"

Figure 8 -> one point is orange while the others are blue, please explain this in the figure caption or correct if an error.

Line 271 –> delete "on the other hand"

Line 296 –> delete "on the other hand"

Figure 10 and 11 -> These figures could be combined, and the south Caspian and Permian basin images could be dropped as these areas are discussed elsewhere.

We prefer to keep these figures as they are for the reasons discussed above.

Line 392 –> delete "on the other hand

**Reviewer 2**

The authors here present preliminary results from the MethaneSat satellite. The satellite is presented as bridging the gap between quantifying total regional fluxes and detection of some point sources, but the authors in this paper focus solely on the point source component of the technology. It is really interesting to see these results, and the satellite imagery displayed in this manuscript is quite compelling. I have several comments before being able to recommend for publication - in particular, more detail of their approach and contextualization of their results is needed. The authors also do not have a strong scientific question or hypothesis they are testing, rather they are mostly showing a demonstration and capability of a new technology. Given that focus, it's important that they do not draw too broad of conclusions based on these paper results alone. That said, I do look forward to what should ultimately be a deep set of scientific analyses that will come from the MethaneSat satellite record.

Thank you for your review of the manuscript and the several useful comments and suggestions. Please, find below our responses to your points.

1. Abstract - I am worried the authors are overstating a few things based on the evidence they've shown in the paper. In particular, Line 15. "Our results illustrate the potential of the MethaneSAT data archive for the discovery of new methane hotspot regions and super-emitters around the planet." This is not supported yet by the results in this paper. All the regions shown in this paper were previously known to be emitting methane super-emitters via other satellite detection. The authors need to reframe accordingly.

Rephrased to "Our results illustrate the potential of MethaneSAT data to map methane emissions from hotspot regions and super-emitters around the planet."

2. How are the plumes actually quantified? Reading through the list of other cited papers (Chan Miller et al. 2024, Guanter et al., 2025) that use MethaneAir data, it appears that either a divergence integral method could have been applied, or an IME method. In the text the authors state "mass balance" but that could apply to variety of approaches. If it's an IME method (or really for either), how did the authors tune parameters to arrive at these emission rates? Through the comparison with IMEO-MARS data? Through simulation? This warrants deeper discussion given the presentation of emission rates in this paper.

In our understanding, no fine-tunning is done for the DI-growing box method developed for MethaneSAT and MethaneAIR.

This text has been added: "In particular, the DI-growing box method used for the quantification of MethaneSAT plumes applies Gauss's divergence theorem to estimate methane emissions by integrating methane flux divergence within expanding boxes that enclose the plume, eliminating the need to define inflow concentrations. By averaging methane gradients with wind vectors around the plume, it is well suited for larger or clustered sources but less sensitive to small point

emitters. This method has been validated by controlled release experiments \citep{ElAbbadi_2024}."

3. Line 283. What makes these plumes unprecedented? That is an overstatement given that all the satellites listed in this paper (GHGSat, EnMAP, EMIT, etc) have seen plumes of the same magnitude or even at lower detection limits in Turkmenistan.

We were meaning an "unprecedented view" of the plumes, not the plumes themself. In thy case, this wording has been rephrased to "*MethaneSAT's XCH$_4$ maps provide a unique view of methane super-emissions in this basin thanks to MethaneSAT's wide coverage and relatively fine spatial resolution.*"

4. Section 3.2.2. Can the authors provide more details about the emission rates distributions and the total point source emission total for all the Permian overpasses? In Figure A1 I only see the 22-May 2024 totals, but then this section is try to draw contrasts with other overpasses to say something about intermittency. Would be interesting to see how much the total point source rate changes across these observations.

A new Fig.A2 has been added to show the plume distributions and totals from all the Permian scenes in this work.

The variability of total point source rate changes is now discussed as: "*This variability can also be seen in the total point source emissions reported in Fig.\,\ref{fig:Q_bars_Permian}: the maximum total point source emission is found on 28 September (87.8\,t/h), which shrinks by a factor \sim{5x} on the next day (16.3\,t/h). Intermediate and more similar values are found on 25 and 25 October (50.4 and 57.6\,t/h).*"

5. In Figure A1, it looks like the total point source estimate is around 25 t/h, though still curious about those totals for other overpasses (comment #4). I crossed reference this against the MethaneAir paper the authors cited (Guanter et al., 2025) that also imaged the Permian and found ~35 t/h. Could you use that result or comparison (or maybe the distributions of MethaneAir vs MethaneSat) to say something more rigorous about MethaneSat's detection limit?

We have addressed this point by a new data analysis, this time including all available plumes in the MethaneSAT archive as suggested by the reviewer. We have fitted a lognormal curve to the histogram of all available plume detections, which is shown in the new Fig. 5. The point at which the curve rolls off toward smaller values (about 1300 kg/h) has been taken as an "average" detection limit for the collection of MethaneSAT datasets contributing to the plume list.

6. In the comparison with other satellite missions, the authors note that there can be grouping of multiple unique plumes into one plume complex, which complicates comparison. Then in section 3.2.2, the authors state that they see some plume emission rates that are exceptionally large

relative to other studies - on Line 315 they state seeing 5 plumes greater than 5 tons/hr. Couldn't this really just be an issue of multiple plume aggregation? Given that repeated remote studies - the authors cite some (MethaneAir, Carbon Mapper, PRISMA/GF5) and miss some others (Insight M; Chen, Sherwin et al. 2021:https://doi.org/10.1021/acs.est.1c06458; Kunkel et al. 2023: https://doi.org/10.1021/acs.est.3c00229) - but none of those other studies see that frequency of exceptionally large emission sources, which calls into question whether the emission rates derived from MethaneSat truly represent individual sources, or whether the quantification is biased in some way. Can the authors further clarify what could be driving the exceptionally large emission rates? Why are you certain there isn't a quantification bias?

We have added the following clarification: "*Considering how clear those plumes appear in the MethaneSAT maps, we have confidence that the high rates that we have estimated are real. Also, zooming in over single plumes, we do not find evidence of the merging of several sources. The fact that many of the plumes are from the same sources on subsequent days with somewhat different patterns in the winds gives us some confidence that the reported plumes are from single sources are not coincidental aggregation of flux from multiple sources.*"

Those large plumes in the Midland basin are actually one of the main findings of this study, as highlighted in the Abstract.

7. Can the authors provide the raw data from Figure A1 (and all plume detections + quantified emission rates) as part of this submission?

Done, thanks for the suggestion.

8. Authors state potential takeaway capacity limitations as driving the emissions they see in the Permian. Given the relatively small number of plume detections, I suggest the authors attribute the plumes to the most plausible source infrastructure types. That should be relatively easy and would be preferred to speculation.

This paragraph has been rewritten as shown below in order to reduce the amount of speculation without data while providing the underlying attributions like the reviewer suggested

 "*Prior research has shown variation in Permian emissions connected with takeaway capacity and concurrent variation in oil prices (Lyon et al., 2021). Before additional pipeline capacity came online in November 2024, west Texas natural gas spot prices were frequently negative throughout the summer and early fall of 2024, creating an environment where Permian operators lost money to transport natural gas (U.S. Energy Information Administration, 2025). We note that 76% (23/30) of the marked plumes (Figure 11) originated from midstream facilities, including compressor stations, processing plants, and gathering pipelines. The heighted rate of plumes from midstream facilities may be indicative of the beforementioned limitations in takeaway capacity for the broader Permian at that time of the year.*"

9. Figures 4b and 4d. Presumably Hassi Messaoud is going to be the optimal condition for plume detection (bright and homogeneous). As the authors point out, to the naked eye, the embedded synthetic methane plume in 4d, is marginally discernible at 1.6 m/s, and disappears at 3 m/s. Related, there is another enhancement in all figures around the index (x=420, y=90) that could potentially be a plume, or is at least as compelling as the LES 500 kg/h plume enhancement in Figure 4b. So this raises the question on how the plume detection process is taking place in the MethaneSat processing system. Can the authors comment on how the build confidence in a plume detection?

Thanks, well spotted that there is a likely plume in the Algeria dataset that has been missed by the L4 processing. We have acknowledged this in the text as "*whereas one likely plume may have been missed around pixel (450, 90)*".

The plume detection algorithm in the L4 processing chain has evolved over time and more advanced methods are in place now.

The plume detection process has been described as: "*Plumes are detected and masked through automatic data processing methods. The first version of these methods relied on methane concentration thresholds to detect and segment the plumes, but more sophisticated methods consisting in data denoising and different criteria for plume detection and masking (including concentration, plume shape and wind direction) were implemented later \citep{zhanz_Wavelet}. The plumes detected by this automated processing are verified by a human before becoming public.*"

10. Figure 5. In the zoomed out view, both of these look like diffusive fields. I do appreciate seeing the whole image, but it's hard to reference the detections shown on the albedo map vs the actual concentration fields that are provided. Related - I think on the 14-June map, the "14-June-2024" label is covering one of the plumes

Thanks. We have explored different ways of representing those maps, but haven't been able to find a better format. In our opinion, the different XCH4 distributions and gradients in the two dates are well captured by the current format of the maps, so we would prefer to keep them as they are.

The 14-June-2024 label has been moved to the center of the map so the plume mentioned by the reviewer is not covered.

11. Line 325. Typo - "four" instead of "fout"

Thanks, corrected

12. The reported 20-30 ppb precision (~1-1.5% of background) is about a factor 2 higher than the single-sounding precision cited for TROPOMI, and is about on par with cited numbers from some point source imaging missions (Ramier et al, 2025:

https://eartharxiv.org/repository/view/9307/; Duren et al. 2025:https://doi.org/10.5194/egusphere-2025-2275), while significantly better than precision numbers from other missions (PRISMA; Guanter et al., 2021). Given the emphasis on MethaneSat filling an area flux and point source imaging gap, and it is worthwhile for the authors to contextualize their precision against other missions.

We think that it is difficult to compare rough precision numbers between instruments with a different spatial sampling and spectral resolution (driving the sensitivity of the retrieval to the surface albedo).

Thls ines have been added: "*For context, high resolution imaging spectrometers being used for plume detection, such as EnMAP or PRISMA, can have similar retrieval precision errors over ideal observation conditions (namely, homogeneous surfaces and high solar irradiance) \citep{javier_enmap_2024}, but the precision errors and the occurrence of false positives increase rapidly with the heterogeneity of the surface due to the coarse spectral sampling of those instruments \citep{guanter_prisma}.*"

13. I understand that this paper's scope is not to map a realistic detection limit using detection probabilities. That said, this is really where the community is going, and minimum detection limit estimation is not really a super helpful metric for understanding the true detection capability of an instrument. I suggest the following that can at least provide more context about how detection limits vary in the areas MethaneSat observed: specifically, can you plot precision (and/or MDL via the MDL equation) either as a function of albedo, or for each of the scenes you summarize in Figure A1? It would be helpful to know how much the precision varies per geography and how that scales the MDL.

Thanks for this comment and suggestion.

We have improved the discussion of retrieval precision vs albedo by extending the discussion of the plume detection drivers. There is now a dedicated section for this topic, "*3.1.2 Assessment of factors driving the detection of methane plumes with MethaneSAT*", and two more datasets have been included in Fig.3 so that it now shows a higher variety of surface types, retrieval precision and wind speed. This text has been added:

"These four acquisitions have been selected in order to sample a wide range of acquisition conditions, in particular of at-sensor radiance (from very low in the Appalachian and West Siberia scenes to very bright in the Permian and Algeria acquisitions) and of wind intensity (from very low in the Appalachian to very strong in the Permian)."

**Reviewer 3**

The paper reports a global survey of methane point hot spots by EDF's MethaneSAT's high spectral resolution (0.25 nm), medium spatial sampling (110×400 m2 at nadir), and wide-area coverage (about 200 km2 at nadir) over its operational lifetime I year 3 months (March 24-June 25). MethaneSAT's ability to identify both point and diffuse large area super-emitting regions from O&G, agriculture and landfills is demonstrated with its L4 product (with citations for details). The paper partially mines MethaneSAT's archive to identify and quantify 16 plumes, with emission rates ranging between roughly 800 and 7000 kg/h. WRF simulations are used to attribute plumes and infer emissions. Results are compared with available IMEO/TROPOMI data sets for rudimentary validation. Given that MethaneSAT was a high-quality NASA/ESA/JAXA-class satellite sensor funded by the non-profit and deployed at a much lower cost this is a critical paper that will help us develop the path forward on future low/medium cost global monitoring of natural gas leaks at a critical time. However, there are important gaps in its methodology and clarity that need to be addressed before I can recommend it for publication.

Thanks to the reviewer for the feedback on our manuscript.

We would like to clarify two points from the above paragraph:

-   "The paper partially mines MethaneSAT's archive to identify and quantify 16 plumes, with emission rates ranging between roughly 800 and 7000 kg/h". This may be referring to the Appalachian scene, which is just one of the several study cases discussed in this work, and not at all the focus of the study.
-   "WRF simulations are used to attribute plumes and infer emissions." We are not sure how WRF simulations could be used to "attribute emissions", and it is certainly not what we do. Also, we do not use WRF simulations to infer emissions in any case (just for a sensitivity analysis of MethaneSAT's plume detection limits.

MeathneSAT's plume detection capability is illustrated and compared with much coarser TROPOMI data using "adjustable arbitrary" parameters – notable "q" that measures the signal/noise at a pixel (# of standard deviations above noise) and np to measure the number of samples above noise threshold. While its ad hoc and qualitative nature are acknowledged no sensitivity studies to these parameters are presented and should be discussed. Their potential impacts on the matched filter plume reconstruction mentioned that use downscaled 30x30m WRF plume simulations that have uncertainties associated with winds and plume dispersion in GEOS-FP compared to the real world. It would also be valuable to MARK the point sources in Fig 1 (a) in addition to 1 (c) to illustrate the fidelity/challenges of plume separation/mixing (as is done later for less cluttered images).

We find several independent comments in this paragraph:

-   "MeathneSAT's plume detection capability is illustrated and compared with much coarser TROPOMI data using "adjustable arbitrary" parameters": Actually, those parameters

were used as part of a model to assess the drivers for the detection of methane plumes with MethaneSAT, they have nothing to do with the comparison with TROPOMI. In any case, we consider that this caveat in the text is still valid: "*we do not interpret $Q_\textrm{min}$ as an absolute measure of the minimum detectable emission rate, but only use it for comparative analysis, This includes the investigation of the relative impact of the different variables driving the plume detection process, and the intercomparison of the plume detection potential among different MethaneSAT L2 datasets.*".

- "Their potential impacts on the matched filter plume reconstruction mentioned that use downscaled 30x30m WRF plume simulations that have uncertainties associated with winds and plume dispersion in GEOS-FP compared to the real world": We are not sure what the reviewer is referring to with this comment. We only used the WRF-LES plume simulations for a rough estimation of MethaneSAT's minimum plume detections in Fig. 6.

- "It would also be valuable to MARK the point sources in Fig 1 (a) in addition to 1 (c) to illustrate the fidelity/challenges": Thanks for this comment. We did try to add the plume detections to Fig.1a as suggested by the reviewer, but finally preferred to keep the figure as is, because this is the first XCH4 map that we show in the document and the plume detection marks did not allow to appreciate the XCH4 gradients in the area.

Detection limits of 500 Kg/h under favorable winds are a key determination/claim that needs more careful caveats and/or evaluation. What are the favorable conditions? What is the limit under unfavorable conditions? What are the uncertainties? Also, standard deviation does not measure "systematic" biases and the sources of this should be mentioned and/or discussed.

We have added the following clarification: "*We would like to note that these results provide a rough estimate of the minimum detection limit of MethaneSAT, which would correspond to very bright and homogeneous areas such as the one in Algeria used for this test, and to low winds as described in Sect.\,\ref{sec:res_det_lim}.*"

There is a lot of discussion in the text on sensitivity/resolving power as a function of wind speed, direction and nonuniformity, surface albedo, cloud cover, multiple sources, matched filter low bias etc… that would be valuable to consolidate in a table for clarity.

We don't have the methods in place to generate those numbers systematically from all the acquisitions shown in this work. We hope that the new Fig. 3, showing now some of those numbers for 4 different acquisitions, can partly represent the information that would be contained in the table mentioned by the reviewer.

The weakest part of the paper is proper uncertainty analysis and validation that demands much more discussion. For example, recent analysis using airborne-AVIRIS ng images of methane releases illustrates how do get reconstruct plume velocities from its shape and use this to show quantitative https://www.pnas.org/doi/10.1073/pnas.2507350122  – This should be cited and

discussed. Are there similar methods that can be developed using WRF and/or MethaneSAT plume information? Can satellites be designed to take multiple snapshots of the plume that are closer in time to allow better velocity constraints for flux inversions?

We are not sure about how to relate the reconstruction of plume velocities from multiple snapshots with the topic of our work, which is the presentation of MethaneSAT's potential and limitations for the monitoring of methane plumes. MethaneSAT can not take such multiple acquisitions within the very short delta time of some minutes which would be need to infer wind speeds from them (and the same likely applies to any other methane-sensitive satellite mission).

Typos and question
Fig 8: Explain why the 'source merged" – yellow L4 is much higher than IMEO
L28 tradeoff
L318 Delaware
L321 Also,
L325 four not fout, cannot
L386 widely used

Thanks, typos corrected and this clarification has been added " The point with the largest deviation between the two datasets is the one from the merged Landsat plumes, for which we can expect the largest uncertainty."